# Regulation of stress granule formation in human oligodendrocytes

Florian Pernin[1], Qiao-Ling Cui[1], Abdulshakour Mohammadnia[1], Milton G. F. Fernandes [1], Jeffery A. Hall[2], Myriam Srour[3], Roy W. R. Dudley[4], Stephanie E. J. Zandee[5], Wendy Klement[5], Alexandre Prat [5], Hannah E. Salapa [6], Michael C. Levin [6], G. R. Wayne Moore[1], Timothy E. Kennedy [1], Christine Vande Velde [5] & Jack P. Antel [1] ✉

Oligodendrocyte (OL) injury and subsequent loss is a pathologic hallmark of multiple sclerosis (MS). Stress granules (SGs) are membrane-less organelles containing mRNAs stalled in translation and considered as participants of the cellular response to stress. Here we show SGs in OLs in active and inactive areas of MS lesions as well as in normal-appearing white matter. In cultures of primary human adult brain derived OLs, metabolic stress conditions induce transient SG formation in these cells. Combining pro-inflammatory cytokines, which alone do not induce SG formation, with metabolic stress results in persistence of SGs. Unlike sodium arsenite, metabolic stress induced SG formation is not blocked by the integrated stress response inhibitor. Glycolytic inhibition also induces persistent SGs indicating the dependence of SG formation and disassembly on the energetic glycolytic properties of human OLs. We conclude that SG persistence in OLs in MS reflects their response to a combination of metabolic stress and pro-inflammatory conditions.

Oligodendrocyte (OL) injury and subsequent loss is a pathologic hallmark of multiple sclerosis (MS), the most frequently acquired demyelinating disorder of the central nervous system. The initial acute demyelinating lesions that feature adaptive and innate immune infiltrates and that are amenable to systemic immune-directed therapies have relative preservation of OLs albeit with dying back of cellular processes. OL cell loss is variable in post-demyelinating active lesions but their loss is consistently found in mixed active-inactive ("chronic active") and chronic inactive lesions; the latter type of lesions is linked to the progressive disease phase that is resistant to systemic immuno-therapies. "Chronic active" lesions feature innate immune cells (macrophage/microglia) at lesion edges, providing a potential source of pro-inflammatory cytokines[1–3]. In addition, histopathologic studies showing an accumulation of metabolic stress-related products and imaging studies indicating reduced perfusion and hypometabolism

implicate oxidative stress and energy failure as contributors to lesion progression[4–8]. RNA sequencing studies have identified "stressed OLs" populations in MS tissues with upregulated cell stress mechanisms including the unfolded protein response (ATF4) and downstream associated chaperones (HSP90AA1)[9,10].

Stress granules (SGs) are complex condensates of messenger ribonucleoprotein that contain messenger ribonucleic acids (mRNAs) stalled in translation initiation[11]. Such formation has been linked with RNA binding proteins (RBPs) and associated with the presence of "stress environments"[12]. Although these structures have been hypothesized to play a protective role, the precise function of SGs remains to be defined. SG assembly and disassembly is a highly regulated and multi-step process linked to pathways that regulate protein translation and clearance. Central to the translation-controlling program are the Integrated Stress Response (ISR) and the mechanistic Target of

[1]Neuroimmunology Unit, Montreal Neurological Institute, McGill University, Montreal, QC, Canada. [2]Department of Neurosurgery, McGill University Health Centre, Montreal, QC, Canada. [3]Division of Pediatric Neurology, Montreal Children's Hospital, Montreal, QC, Canada. [4]Department of Pediatric Neurosurgery, Montreal Children's Hospital, Montreal, QC, Canada. [5]Centre de Recherche Hospitalier de l'Université de Montréal, Montréal, QC, Canada. [6]Cameco Multiple Sclerosis Neuroscience Research Center, University of Saskatchewan, Saskatoon, SK, Canada. ✉e-mail: jack.antel@mcgill.ca

Rapamycin (mTOR) pathways[13]; their interactions play a crucial role in reprogramming the expression of genes that function to adapt cells to stress[14]. Canonical SG formation is usually associated with the phosphorylation of eIF2α and downstream ISR activation[15]. We previously demonstrated that the core signaling ISR pathway is activated in injured OLs in MS active lesions, as shown by a significant increase in p-eIF2α[16], a key regulator of the ISR pathway. However, recent findings demonstrate alternative mechanisms that are p-eIF2α independent, acting on the nutrient sensor mTORC1[17,18]. In this context, mTORC1 specifically drives the eIF4E-mediated formation of SGs through the phosphorylation of 4E-BP1, a key factor known to inhibit the formation of mTORC1-dependent eIF4E-eIF4G interactions. Salapa and colleagues have previously provided evidence of SG dysfunction that may underlie neuronal damage in MS disorder[19].

The current paper aimed to evaluate the presence of SGs in OLs as an indicator of injury in situ in MS tissues and use dissociated cultures of human brain-derived OLs to define the mechanisms that regulate the development and persistence of SGs in response to acute stress (sodium arsenite, SA) or chronic metabolic stress (glucose/nutrient deprivation) and inflammatory cytokines (TNFα, IFNγ). We document the presence of SGs in areas of active MS lesions, even in regions without significant OL cell loss, as well as SG persistence in inactive lesion areas where OL numbers are consistently reduced, and in normal appearing white (NAWM). As evidence of concurrent stress conditions, we demonstrate increased expression of ATF4, a master regulatory protein of the integrated stress response (ISR) pathway in OLs in active MS lesions while expression of the mTOR pathway marker p4E-BP1 is reduced. In our in vitro studies of human primary OLs, we show distinct differences in the participation of the ISR and mTOR protein regulatory pathways in SG assembly under acute compared to chronic metabolic stress conditions. SA induces persistent SG formation that is inhibited by pharmacologic blockade of the ISR. Metabolic stress induces transient expression of SGs that is not inhibited by the ISR inhibitor (ISRIB). Application of the mTOR inhibitor Torin1 results in persistent SG formation under both metabolic stress and control conditions. Pro-inflammatory cytokines themselves do not induce SG formation. However, combining the latter with metabolic stress conditions results in SG persistence in the OLs, suggesting a failure for timely dissolution of SGs. We further show the dependence of SG persistence on the glycolytic metabolic properties of the OLs. We suggest that the persistence of SGs in OLs in MS reflects distinct changes in the protein translation regulatory pathways and glycolytic metabolic properties of these cells in response to a combination of metabolic stress and pro-inflammatory conditions.

## Results

### In situ studies—SG formation participates in the OLs injury response in MS

We first compared the expression of SGs in OLs in post-mortem tissue samples from MS patients and controls (Ctrl). Patients characteristics are shown in the supplementary material (Table S1). MS lesion activity was characterized by Luxol Fast Blue (LFB) and Hematoxylin and Eosin (H&E) staining. Areas of ongoing demyelination were selected based on the presence of macrophages/microglia, some of which contained LFB-positive material (Fig. S1A). There were also areas of inactivity in these active/mixed lesions. We determined the number of OLs in MS tissue sections using the mature OL marker NogoA. Quantitative analyses revealed no significant differences in OL cell numbers between the NAWM regions of MS patients versus control white matter (Fig. 1A). We quantified a slight decrease in NogoA+ cells in active areas of MS lesions compared to NAWM regions (adjacent or distal). As reported by Heß et al. [20], a major loss of OL cell number was observed in the inactive MS lesion areas (Fig. 1A).

To detect SGs, we performed immunohistochemistry for two markers of SG formation, PABP and G3BP1. Both markers were diffusely distributed in OLs in control brain tissue samples (Fig. 1B, C and Fig. S1B, C). SGs were present in the large majority of the OL population (NogoA+ cells) in both the active areas of MS lesions as well as in the NAWM (distal and adjacent). SGs were also found in the remaining OLs in the inactive lesion. As shown in Fig. S2A, we can also detect SG formation in astrocytes within MS lesions.

### Translation-controlling pathways (ISR and mTOR) are implicated in OLs injury response in MS lesions

We previously showed the increased phosphorylation of the core molecule of the ISR pathway, eIF2α, in MS lesions[16]. Here, we expanded this initial evaluation by assessing the expression of ATF4 and 4E-BP1 phosphorylation levels using quantitative IHC. In the active areas of MS lesions, we observed a significant increase in the proportion of OLs expressing high levels of ATF4 (49% ± 4.9) compared to control (7% ± 4) (Fig. 1D, E). Only 19% ± 13.4% of OLs in active lesion areas have high expression of phosphorylated 4E-BP1. In contrast, control patients demonstrate sustained mTOR activity and protein synthesis, as shown by a high expression of p4E-BP1 (85% ± 3.5) (Fig. 1F, G). ATF4 was reduced in the inactive areas of MS lesions compared to active regions, whereas p4E-BP1 in the inactive areas remains reduced compared to the control. These findings suggest an ongoing stress response in the active lesion areas (high ISR and low mTOR activity) leading to SG formation; persistence in inactive areas of MS lesions that lack an ongoing significant stress response (low ISR and low mTOR activity) would reflect a failure of requirements for SG disassembly. NAWM regions of the MS tissue samples showed an intermediate expression profile of the ATF4 and p4E-BP1 (Fig. 1D–G). The presence of SGs in NAWM implicates ongoing OL injury that may be independent of plaques and potentially represent pre-lesional activity[21].

### In vitro studies—differential SG dynamics in human primary OLs exposed to injury-inducing conditions

For these studies, we exposed human primary mature OLs (hOLs) isolated from surgically resected brain tissue samples to the following conditions: i) established SG-provoking reagent, sodium arsenite (SA) considered as an acute type of oxidative stress condition; ii) conditions implicated in the ongoing (chronic) injury process in MS, namely metabolic stress (low or no glucose/nutrient, LG/NG), known to initially induce sublethal injury within 48 h, followed by cell death over a 4–6 day time period; and iii) pro-inflammatory cytokines shown to induce sublethal injury (process retraction). In these in vitro studies, SGs were detected by immunostaining using G3BP1, which is a unique scaffolding protein for SG assembly.

We first assessed the dynamic nature of SG assembly and disassembly over 4 days in hOLs treated with SA. SGs were easily visible in hOLs exposed to SA conditions, whereas control conditions showed diffuse G3BP1 staining (Fig. 2A, B). We found that the formation of SA-induced SGs started within 30 min after exposure. These granules persisted during the injury response of hOLs until the cell viability was significantly compromised (Fig. 2C).

On exposure to SA, hOLs activated the programmed cell death pathway, apoptosis, as shown by the increased level of Casp3/7 staining in the population (Fig. S3A). We have previously shown that hOLs treated with other types of stress conditions are resistant to pro-apoptotic cell death[22].

Because canonical SGs disassemble following the removal of the initial stressor, we examined whether removing SA and allowing cells to recover might decrease the presence of SG-positive cells. We observed that SGs rapidly resolved after the withdrawal of SA. Fewer than 50% of OL population displayed SGs after 4 h post-SA exposure (Fig. S3B). There was no increase in subsequent cell death if cells were exposed to SA for only 1 h (3.7% ± 2.2 for control vs 5.2% ± 2.9 for SA).

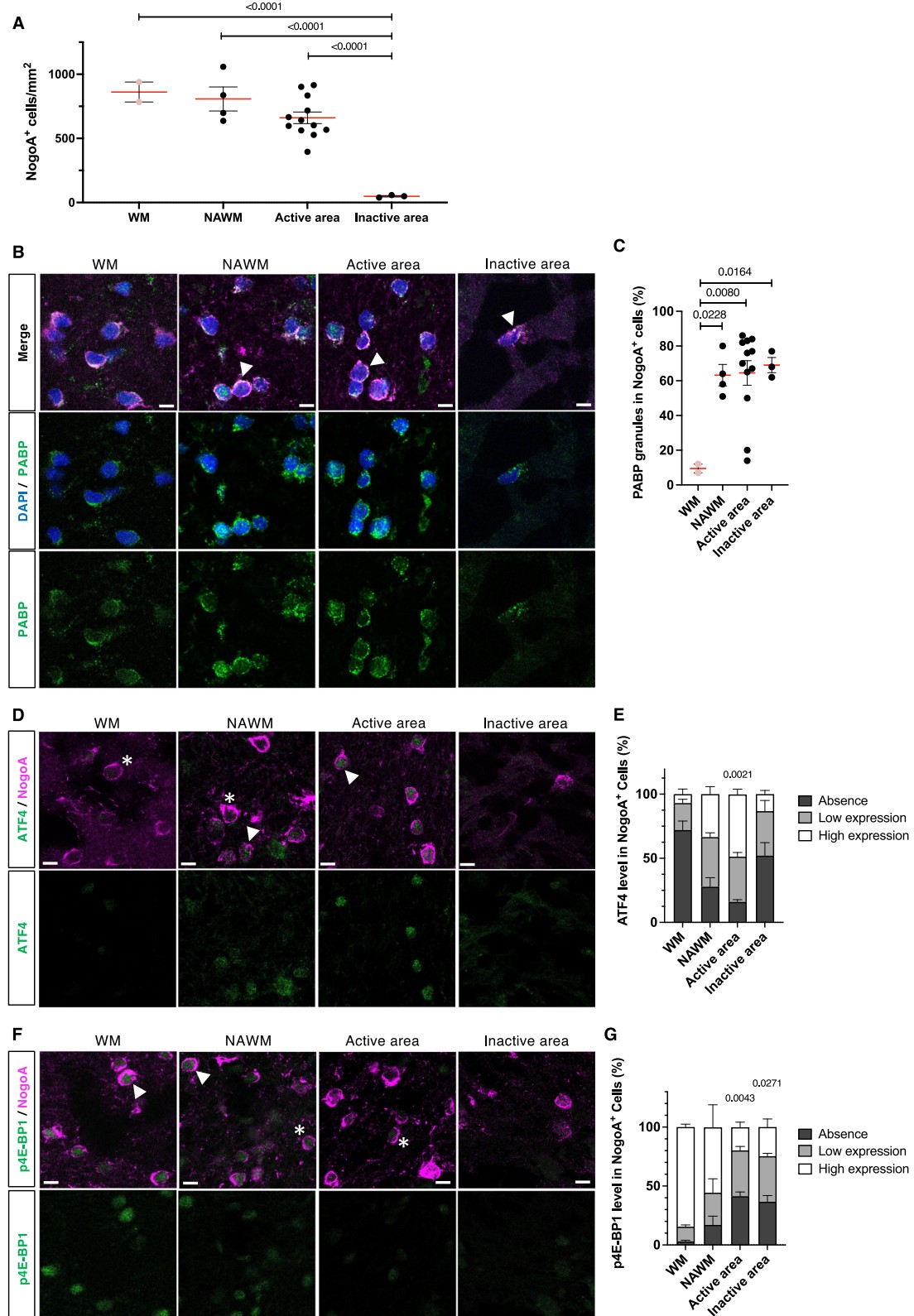

## Metabolic stress conditions induce transient SG formation

In contrast, hOLs exposed to NG conditions showed a rapid and transient formation of SGs, peaking at 4 h. At this time, SGs were visible in approximately 60% of the OL population (Fig. 2A, B). Low glucose, a milder metabolic stress condition (see Fig. 4 below), was not sufficient to trigger the formation of SGs in hOLs at any time of the injury response.

## Pro-inflammatory conditions do not induce SGs but in combination with metabolic stress result in persistent SG formation

Pro-inflammatory cytokines are implicated in perpetuating tissue injury in MS[23–25]. We have previously shown that TNFα and IFNγ induce process retraction in hOLs in vitro with a limited increase in cell death[16]. To test if these stress mediators could induce SG formation,

**Fig. 1 | SG formation as an indicator of OLs injury in MS lesion. A** Quantification of OLs in control tissue (WM) and different brain regions of MS cases using NogoA as a marker for mature OLs. **B** Representative confocal images of IHC staining for stress granules (PABP⁺) in OLs different brain regions of control and MS cases. Note the normal diffuse PABP staining in control (WM) OLs and the granular staining in MS OLs indicative of SGs. PABP granules are predominantly formed in the cytoplasm of OLs (arrowheads). **C** Quantification and gradient distribution of PABP⁺ OLs were assessed in control and MS tissues. Data are graphed as the percentage of OLs positive for PABP granules where the denominator is the number of OLs counted in that region. (**D-G**) Representative confocal images of IHC staining for ATF4 (**D**) and p4E-BP1 (**F**) markers in OLs in control and MS brain regions. Quantification and gradient distribution of these markers in OLs are expressed as a percentage of each category (**E&G**). Statistical analysis was performed on the high expression category, comparing the different MS areas with the controls. Examples of low (stars) and high (arrowheads) expression categories in OLs for each marker are provided in (**D**) and (**F**). Note that DAPI staining was intentionally omitted for better visualization of the ATF4 and p4E-BP1 markers. Scale bars, 10 µm. Sections were stained with DAPI (blue), NogoA (pink) and the marker of interest (green). Analyses done on two control patients accounting for independent regions as WM ($n = 2$) and three MS patients accounting for NAWM ($n = 4$), active ($n = 12$) and inactive ($n = 3$) areas. For each quantification, >100 cells were assessed in each independent area when possible. Each dot in the graphs represents a value from a distinct and individual area. All data are expressed as mean values ± SEM, analyzed by one-way ANOVA followed by Bonferroni's multiple comparisons correction. All significant $P$ values are indicated; ns or unlabeled not significant. WM white matter, NAWM normal appearing white matter, Active area active areas of active and mixed MS lesion, Inactive area inactive area of mixed MS lesion. Source data are provided as a Source Data file.

we treated hOLs with TNFα and IFNγ alone and also in combination with metabolic stress (LG or NG) conditions.

When cells were cultured with pro-inflammatory (TNFα and/or IFNγ) stressors alone or in combination in optimal (ctrl) or mild stress conditions (LG), no SGs were detected by G3BP1 labeling in the early (4 h) or late (24 h) time points (Fig. 2D, E). However, the combination of metabolic stress (NG) conditions with pro-inflammatory cytokines (TNFα and IFNγ) produced a different SG dynamic. Under NG + TNFα + IFNγ conditions, SG formation in hOLs at 4 h of treatment was within a similar range for NG treatment alone (Fig. 2D). However, unlike NG conditions alone, cells under NG conditions supplemented with pro-inflammatory cytokines showed persistent SGs following 24 h of treatment (Fig. 2D, E). This combination effect did not induce an increase in cell death at 24 h, as shown by PI staining compared to NG condition alone (Fig. 2F). These observations suggest that pro-inflammatory cytokines (TNFα and IFNγ) impaired SG disassembly mechanisms and lead to SG persistency in hOLs co-treated with metabolic stress (NG) conditions.

### Comparison of mechanisms underlying SG formation under acute and chronic stress conditions

SG formation has been linked with ATP levels, protein synthesis, and polysome equilibrium under acute and chronic stress conditions in a variety of cell lines[26–28]. Furthermore, the translation-controlling program (the ISR and mTOR pathways) is known to drive the formation of SG in response to stress mediators[29,30]. We thus have investigated the cellular and molecular properties of our hOLs under acute and chronic stress conditions.

### SG formation in hOLs is associated with ATP depletion and translational arrest

SA quickly reduces ATP levels compared to cells in control media (Fig. 3A). The decline of ATP levels continued over time, up to 24 h where increased cell death can be seen (Fig. 2C). NG conditions also induced a significant reduction in ATP level in hOLs starting at 4 h. We then assessed whether translation was repressed under our stress conditions. We measured newly synthesized proteins in SA and NG conditions using an immunofluorescent-based protein synthesis assay (Click-iT). Control hOLs showed high-intensity signals indicative of normal translation (Fig. 3B, C). We observed a significant inhibition of protein synthesis in SA conditions as well as in the positive control where hOLs were treated with the translational elongation inhibitor cycloheximide (Fig. 3B). At 4 h of metabolic stress (NG) conditions, the point at which more than half of the population contains SGs and concomitant with ATP decline, hOLs showed a reduction in global translation (low-intensity signal). We did not observe an effect of the TNFα + IFNγ cytokine combination on ATP or protein synthesis under control conditions and no synergistic effect under NG conditions (Fig. 3A, B).

To demonstrate that the SG-like cytoplasmic foci we observe are actually dynamic, we applied puromycin to our cultures under NG condition at 4 h, the point at which more than half of the population contains SGs (Fig. 3D). This resulted in increased SG formation supporting previous mechanistic findings showing that polysome destabilization with puromycin enhances SG formation during stress. In contrast, adding cycloheximide (CHX), known to stabilize polysome complexes and prevent SG assembly, and perhaps dissolving existing ones, inhibited SG formation in OLs under NG conditions at 4 h and under SA conditions at 24 h. In presence of CHX, SGs were not persistent under NG + TNFα + IFNγ conditions at 24 h (Fig. 3D).

### The formation of SGs in response to metabolic stress conditions involves a phosphorylated-eIF2α independent mechanism

We first assessed the status of the ISR and mTOR pathways, major components of the translation-controlling program and known to be involved in SG formation in a stress-type dependent context[31]. Consistent with the literature, SA treatment induced a significant increase in the phosphorylation level of eIF2α early in the injury response; this persisted for the duration of the survival time of the cells (2 days) (Fig. 4A, B). Consistent with our previous report[16], chronic metabolic stress (NG) conditions showed similar activation of the ISR over time, although less marked compared to SA treatment, as assessed by eIF2α phosphorylation (Fig. 4A). Similarly, cells exposed to SA conditions showed a significant decrease in 4E-BP1 phosphorylation, a measure of mTORC1 activity (Fig. 4C, D). We documented that NG conditions induced a reduction in p4E-BP1 over time, but were less marked as compared to SA treatment. It should be noted that we previously showed that pro-inflammatory cytokines, IFNγ and TNFα, do not induce ISR activation[16].

We next tested if ISR activation drives the dynamics of SGs in hOLs. We used Sephin1 and ISRIB, two molecules acting as agonists and antagonists of the ISR pathway, respectively. Under optimal conditions alone or in the presence of TNFα and IFNγ, Sephin1 alone was not sufficient to induce SG assembly (Fig. S4A). We could not measure an additive effect of Sephin1 on SA-induced SG formation as likely a maximum expression was already present. We found that ISRIB strongly mitigated the formation of SA-induced SGs over time (Fig. 4E). These results established that the canonical SG dynamics (as established by SA exposure) in hOLs is mediated by the activation of the ISR pathway, as previously stated[32]. In contrast, the formation of SGs due to 4 h of NG conditions with or without co-presence of IFNγ and TNFα, was unaffected by Sephin1 or ISRIB (Figs. 4F and S4B).

To test the hypothesis that the mTOR pathway has a direct impact on SG dynamics in hOLs, we used an active site inhibitor of mTORC1 and mTORC2, Torin1. The latter reduces mTORC1 activity i.e., the downstream phosphorylation of 4E-BP1, under control and NG conditions (Fig. 4G, H) with no apparent effect on cell death at 24 h

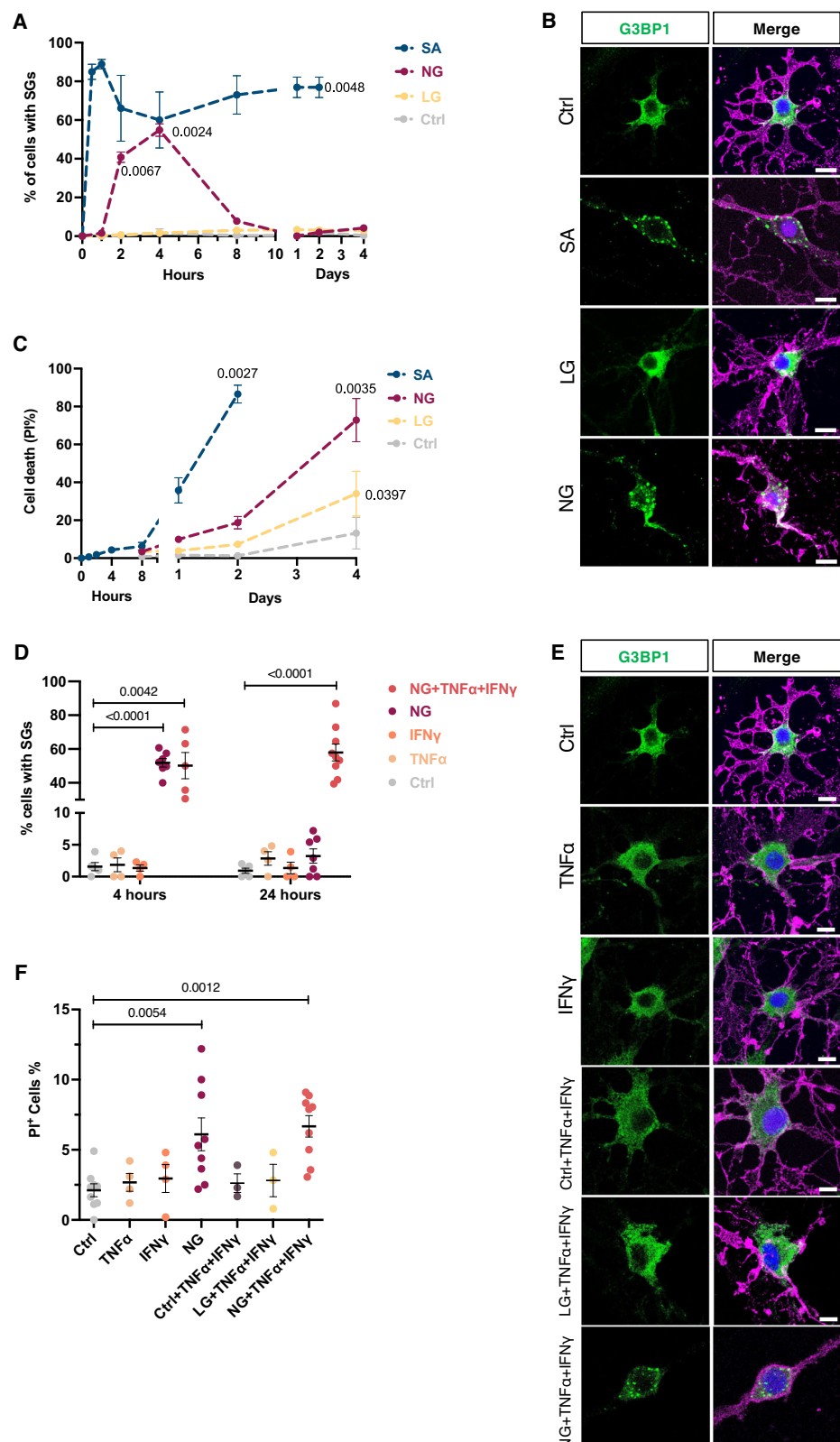

(2.9% ± 3.1 for control vs 4.1% ± 2.2 with Torin1; 5.8% ± 2.7 for NG vs 6.4% ± 3.3 with Torin1).

The addition of Torin1 resulted in SG formation even under optimal conditions ± TNFα + IFNγ (Fig. S4A), as well as SG persistency under NG conditions (Fig. 4I). Our results indicate that persistence of SGs reflects a reduced mTOR activity below a crucial threshold.

## Glycolytic inhibition causes SGs to persist through ATP deficiency

We have previously reported that human primary mature OLs primarily utilize glycolysis for the majority of their energy supply instead of OXPHOS[33]. Recent studies highlighted the importance of this energy source in SG assembly and disassembly mechanisms[28,34]. For this reason, we examined the influence of glycolysis on SG dynamics in NG

**Fig. 2 | Dynamics of SG formation under acute and chronic stress conditions in hOLs. A** Quantification of the percentage of cells with SGs at different time points during continual exposure to Ctrl, SA, LG, or NG conditions. The left part of the X axis shows up to 10 h—the right part of the X axis from 1 to 4 days of treatment (n = 4). **B** Representative confocal images of SG formation in hOLs labeled for G3BP1. hOLs were exposed to the indicated treatments for 4 h. **C** Quantitative analysis of cell death (PI⁺) of hOLs over time. The left part of the X axis shows up to 8 h—the right part of the X axis from 1 to 4 days under the indicated treatments (n = 5). **D** Quantification of the percentage of cells with SGs during continual exposure to Ctrl, TNFα or IFNγ alone or in combination with NG conditions for 4 and 24 h. For each condition, the specific number of biologic samples tested is shown in the graph. NG treatment alone is added here as a comparator. **E** Representative confocal images of SG formation in hOLs labeled for G3BP1. hOLs

were exposed to the indicated treatments for 24 h. **F** Quantitative analysis of cell death (PI⁺) of hOLs at 24 h under the indicated treatment. For each condition, the specific number of biologic samples tested is shown in the graph. NG treatment alone is added here as a comparator. No significant differences were found between NG and NG + TNFα + IFNγ conditions. Scale bars, 10 μm. Merge pictures display DAPI (blue), O4 (pink), and G3BP1 (green). G3BP1 was used as the marker of reference for SG formation. Each dot in the graphs corresponds to an independent biological replicate. All data are expressed as mean values ± SEM, analyzed by paired or unpaired two-tailed Student's *t* test (**A**, **C**) or by one-way ANOVA (**D**, **F**) followed by Bonferroni's multiple comparisons correction. All significant *P* values are indicated, ns or unlabeled not significant. Ctrl optimal media, SA sodium arsenite, LG/NG low or no glucose. Source data are provided as a Source Data file.

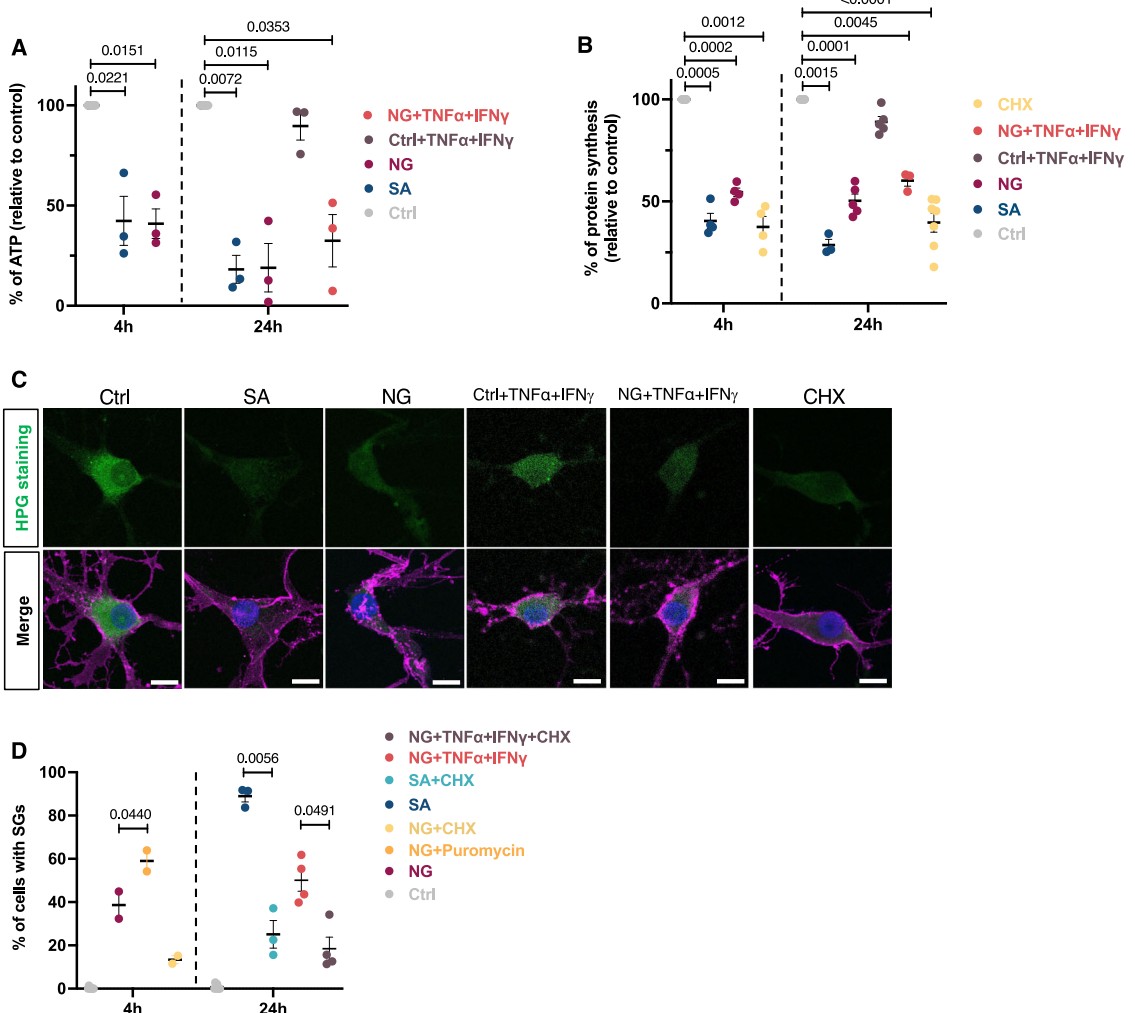

**Fig. 3 | Relationship between SGs and ATP levels and protein synthesis under acute and chronic stress conditions. A** ATP levels in hOLs treated with Ctrl, SA, or NG conditions at different time points. The luminescence (ATP) of each condition was normalized per cell (n = 3). No significant differences were found between NG and NG + TNFα + IFNγ conditions. **B, C** Quantitative analysis of protein synthesis of hOLs treated with Ctrl, SA, NG, and CHX at the indicated time. Protein synthesis was assessed by immunofluorescent signal intensity following labeling with L-homopropargylglycine (HPG) and Click-iT assay kit. Representative confocal images of protein synthesis in hOLs using HPG staining are shown in (**C**). Cells were cultured under Ctrl or stress conditions for 4 h. For each condition, the specific number of biological samples tested is shown in the graph. Control cells show high

signal intensity (normal translation) whereas cells under stress conditions show a significant decrease in signal intensity (translation inhibition). Scale bars, 10 μm. Merge pictures display DAPI (blue), O4 (pink), and HPG (green). (**D**) Quantification of the percentage of cells with SGs during continual exposure to Ctrl, NG, SA, and NG + TNFα + IFNγ conditions in combination with puromycin or CHX for 4 and 24 h. For each condition, the specific number of biological samples tested is shown in the graph. Each dot in the graphs corresponds to an independent biological replicate. All data are expressed as mean values ± SEM, analyzed by one-way ANOVA followed by Bonferroni's multiple comparisons correction. All significant *P* values are indicated; ns or unlabeled not significant. Ctrl optimal media, SA sodium arsenite, NG no glucose, CHX cycloheximide. Source data are provided as a Source Data file.

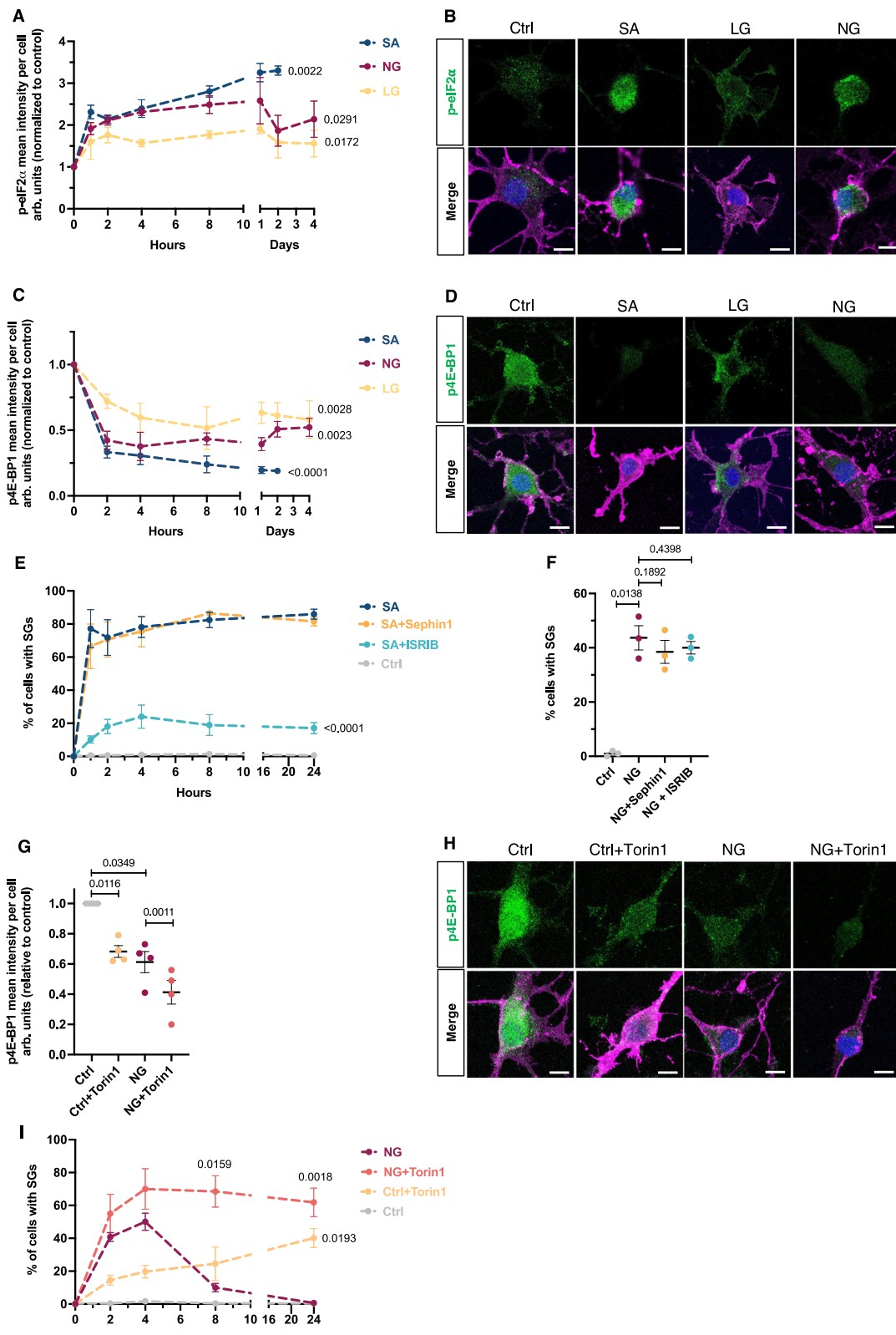

stress conditions. We used two small molecules to block the glycolysis pathway in hOLs: 2-deoxy-D-glucose (2DG), a competitive inhibitor that blocks glycolysis fueled by both extracellular glucose and glycogen[35] and CP91149 (CP), a glycogen phosphorylase inhibitor that shuts down glycogen metabolism[36].

We found that SGs formed and persisted in hOLs exposed to NG conditions when cells were co-treated with the overall glycolysis inhibitor 2-DG (Fig. 5A). SG formation also persisted when treated with the cell-intrinsic glycogen-specific inhibitor CP.

Adding 2-DG or CP inhibitors to cells cultured under optimal conditions (Ctrl) decreased ATP levels by ~30% but induced only a suggestive increase in SG formation compared to Ctrl alone (Fig. 5A, B). Our findings support that the glycolytic pathway is an important component of SG persistence.

**Fig. 4 | Relationship between SG formation and the translation-controlling pathways. A–D** Protein expression profile of the ISR and mTOR pathways, in hOLs. Time course experiment of hOLs treated in vitro with SA, LG, or NG conditions. Cells were labeled with p-eIF2α (**A**) ($n = 3$ independent biological replicates) and p4E-BP1 (**C**) ($n = 5$ independent biological replicates) antibodies. The intensity of these markers was quantified per cell and normalized to the control condition. The left part of the X axis shows up to 10 h—the right part of the X axis from 1 to 4 days under the indicated treatments. Representative confocal images of hOLs are shown for p-eIF2α (**B**) and p4E-BP1 (**D**) at 4 h. **E** Time course experiment of hOLs exposed to SA conditions, in the presence or absence of Sephin1 or ISRIB ($n = 5$). Statistical analysis was performed between SA and SA+drug groups. **F** Quantitative analysis of hOLs exposed to NG conditions, in the presence or absence of Sephin1 or ISRIB for 4 h ($n = 3$). **G, H** Expression profile of p4E-BP1 protein in hOLs exposed to Ctrl or NG conditions in the presence or absence of Torin1 inhibitor for 4 h ($n = 4$). The

intensity of these markers was quantified per cell and normalized to control condition. Representative confocal images of cells after 4 h are shown in (**H**). **I** Time course experiment showing the percentages of SG-positive hOLs, when exposed to NG conditions, in the presence or absence of Torin1 inhibitor ($n = 4$). Statistical analyses were performed between Ctrl and Ctrl+drug or NG and NG + drug groups. Scale bars, 10 μm. Merge pictures display DAPI (blue), O4 (pink), and the marker of interest (green). The percentages of SG-positive cells were assessed by G3BP1. Graphs display arbitrary units (arb. units). Each dot in the graphs corresponds to an independent biological replicate. All data are expressed as mean values ± SEM, analyzed by paired two-tailed Student's $t$ test (**A, C, E, I**) or by one-way ANOVA (**F, G**) followed by Bonferroni's multiple comparisons correction. All significant $P$ values are indicated; ns or unlabeled not significant. Ctrl optimal media, SA sodium arsenite, LG/NG low or no glucose. Source data are provided as a Source Data file.

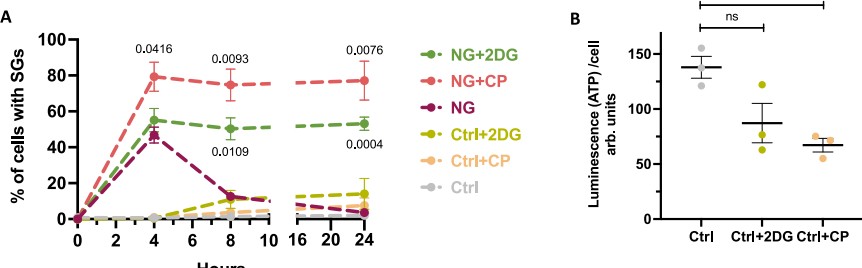

**Fig. 5 | Relationship between SG formation and glycolytic properties of hOLs. A** Time course experiment showing hOLs exposed to Ctrl or NG conditions, in the presence or absence of 2DG ($n = 4$ independent biological replicates) or CP ($n = 4$ independent biological replicates). The percentages of SG-positive cells were assessed by G3BP1. Statistical differences are shown here between NG and NG+drug groups. **B** ATP levels in hOLs cultured for 4 h in Ctrl or NG conditions with or without 2-DG or CP agents. The luminescence (ATP) of each condition was

normalized per cell ($n = 3$). Graphs displays arbitrary units (arb. units). Each dot in the graphs corresponds to an independent biological replicate. Statistical analyses were performed between Ctrl and Ctrl+drug or NG and NG+drug groups. All data are expressed as mean values ± SEM, analyzed by one-way ANOVA followed by Bonferroni's multiple comparisons correction. All significant P-values are indicated; ns or unlabeled not significant. Ctrl optimal media, NG no glucose. Source data are provided as a Source Data file.

## SGs induced by metabolic stress conditions contain multiple SG markers, including PABP and hnRNP A1, but not TDP-43

Many studies have demonstrated that SG formation is tightly linked with RBP biology. We performed immunofluorescence analysis of hOLs in vitro using PABP, hnRNP A1, and TDP-43 antibodies and assessed their co-localization with SGs. In control hOLs, PABP protein displayed a diffuse expression in the cytoplasm of hOLs. Under NG and SA conditions, we visualized PABP granules that co-localized with the SG marker, G3BP1 (Fig. 6A).

In control conditions, hnRNP A1 and TDP-43 and proteins were localized in the nucleus of hOLs (Fig. 6B, C). Following exposure to SA or NG conditions, both, hnRNP A1 and TDP-43 proteins were consistently mislocalized (and also decreased in expression) from the nucleus of the cells at 4 and 24 h. When SGs were induced by exposure of the cells to SA, neither hnRNP A1 nor TDP-43 were recruited to the resultant SGs labeled with G3BP1 (Fig. 6B, C). However, interestingly, we observed that hnRNP A1 and TDP-43 formed aggregates independent of SGs, consistent with previous studies[37,38]. Under NG conditions, we could detect SGs (G3BP1⁺) with and without hnRNP A1 co-localization (Fig. 6B). The latter suggests that this specific RBP is not part of the assembly mechanism but sequestered later during the process. No colocalization was found with TDP-43 (Fig. 6C).

## Molecular insights into SG biology revealed by RNA sequencing analysis

Although the main mechanism of SG formation occurs at the post-translational level, we considered whether the transcription of molecules contributing to SG biology may be impacted in OLs in the MS lesion micro-environment and/or in vitro under our metabolic or pro-inflammatory stress conditions. We used a list of molecules isolated from SGs from Hela cells and characterized by mass spectrometry[28].

This analysis revealed 923 proteins under several types of stress, including sodium arsenite and glucose deprivation conditions (Source Data file).

## Molecular analysis indicates an increased expression of SG-related genes in MS patients

We used publicly available datasets from three recent RNA-seq studies[9,10,39] to investigate the expression of SG-associated genes in OL population in MS cases. We assessed RNA-seq datasets using a single sample gene set enrichment analysis (ssGSEA) and found a similar trend of increased expression of SG-associated genes in MS cases compared to controls in each of the three studies (Fig. 7A–C). We used the Absinta et al MS dataset to analyze the expression profile of SG-associated genes in OLs across lesion pathological stages, including chronic active and chronic inactive regions. In these areas, "stressed OLs" were initially characterized by increased expression of stress responses that included the unfolded protein response, oxidoreductase activity and regulation of cell death. As shown in the Venn diagram in Fig. 7D, the majority of the significantly upregulated genes in chronic active areas were similarly upregulated in chronic inactive areas. The associated bubble plot presents the 32 significantly upregulated shared genes compared to controls (Fig. 7E; respective genes in Fig. S5A). Known functions of these genes include RNA-binding activity involving mRNA processing, transport, and modification (*FXR1*, *hnRNPA2B1*, and *YTHDC1*)[40–42]. We also identified enhanced expression of genes related to chaperone activity (*CALR*, *FKBP4*, *HSP9OAB1*, and *PTGES3*) or autophagy (*SQSTM1*, *RPS27A*), which have been demonstrated to be involved in SG disassembly[27,30,43,44]. Taken together, these results further support our findings of SG formation at the protein level (IHC technique) and show an increased expression of SG-related genes in situ in MS patients.

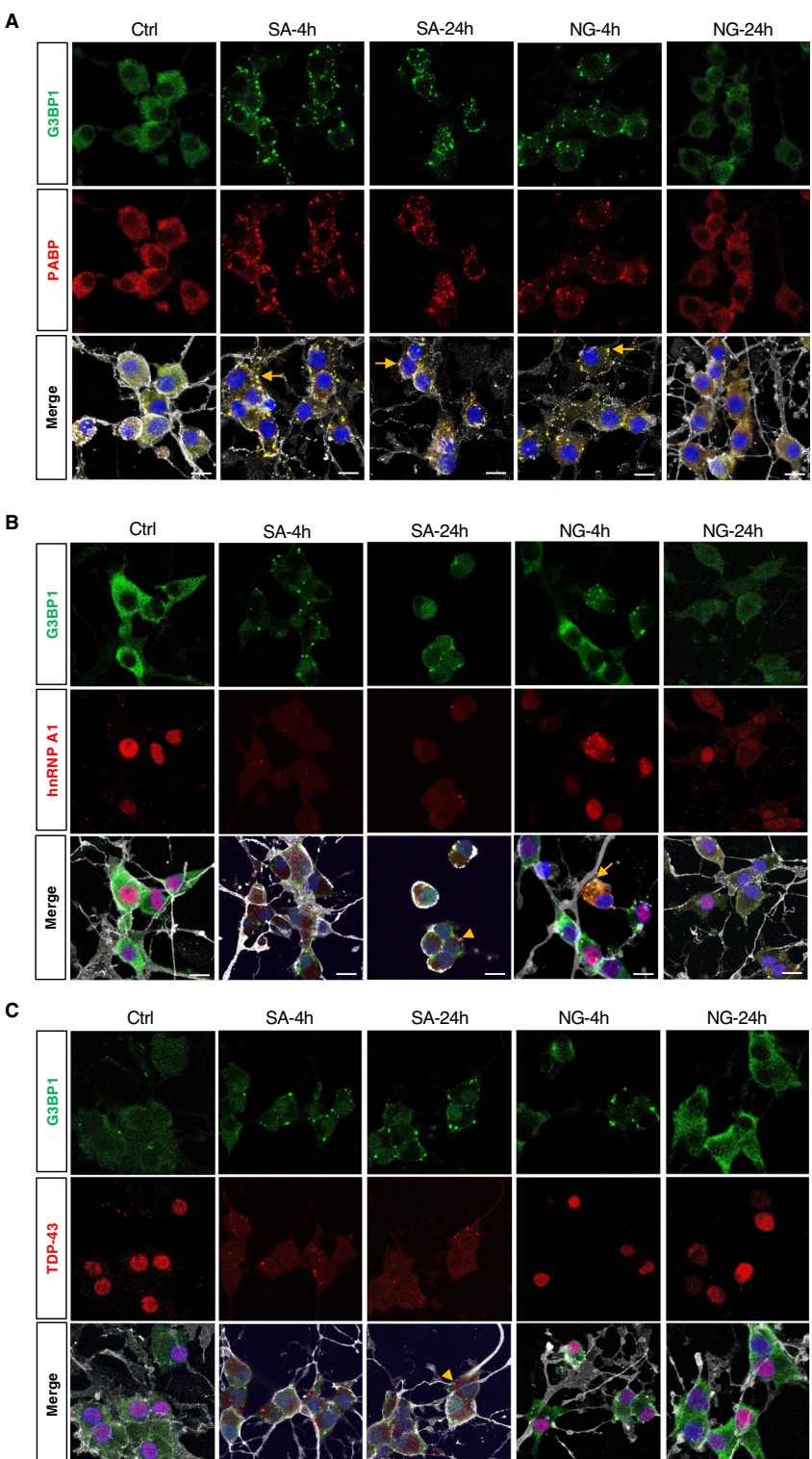

**Fig. 6 | Colocalization of SG markers with RNA-binding proteins.** hOLs were exposed to SA or NG conditions for the indicated periods of time and stained with antibodies recognizing G3BP1 (SG protein component) and RBPs, including PABP, hnRNP A1, and TDP-43. **A** Representative confocal images showing colocalization of PABP with G3BP1 in cells exposed to SA or NG conditions. **B** Representative confocal images showing mislocalization (and some decreased expression) of hnRNP A1 in hOLs and some colocalization with G3BP1⁺ granules at 4 h under NG conditions. **C** Representative confocal images showing mislocalization (and some decreased expression) of TDP-43 in hOLs but no colocalization with G3BP1⁺

granules at 4 h under NG conditions. Treatment with SA resulted in mislocalization (and some decreased expression) of hnRNP A1 (**B**) and TDP-43 (**C**) with the formation of aggregates independent of G3BP1⁺ SGs (arrowheads). Scale bars, 10 μm. Merge pictures display DAPI (blue), O4 (gray), G3BP1 (green) and the marker of interest (red). G3BP1 was used as the marker of reference for SG formation. Arrows indicate colocalization between G3BP1 and the RBP of interest. For each RBP, the colocalization experiments have been repeated 3 times in biologically independent samples, showing similar results. Ctrl optimal media, SA sodium arsenite, NG no glucose.

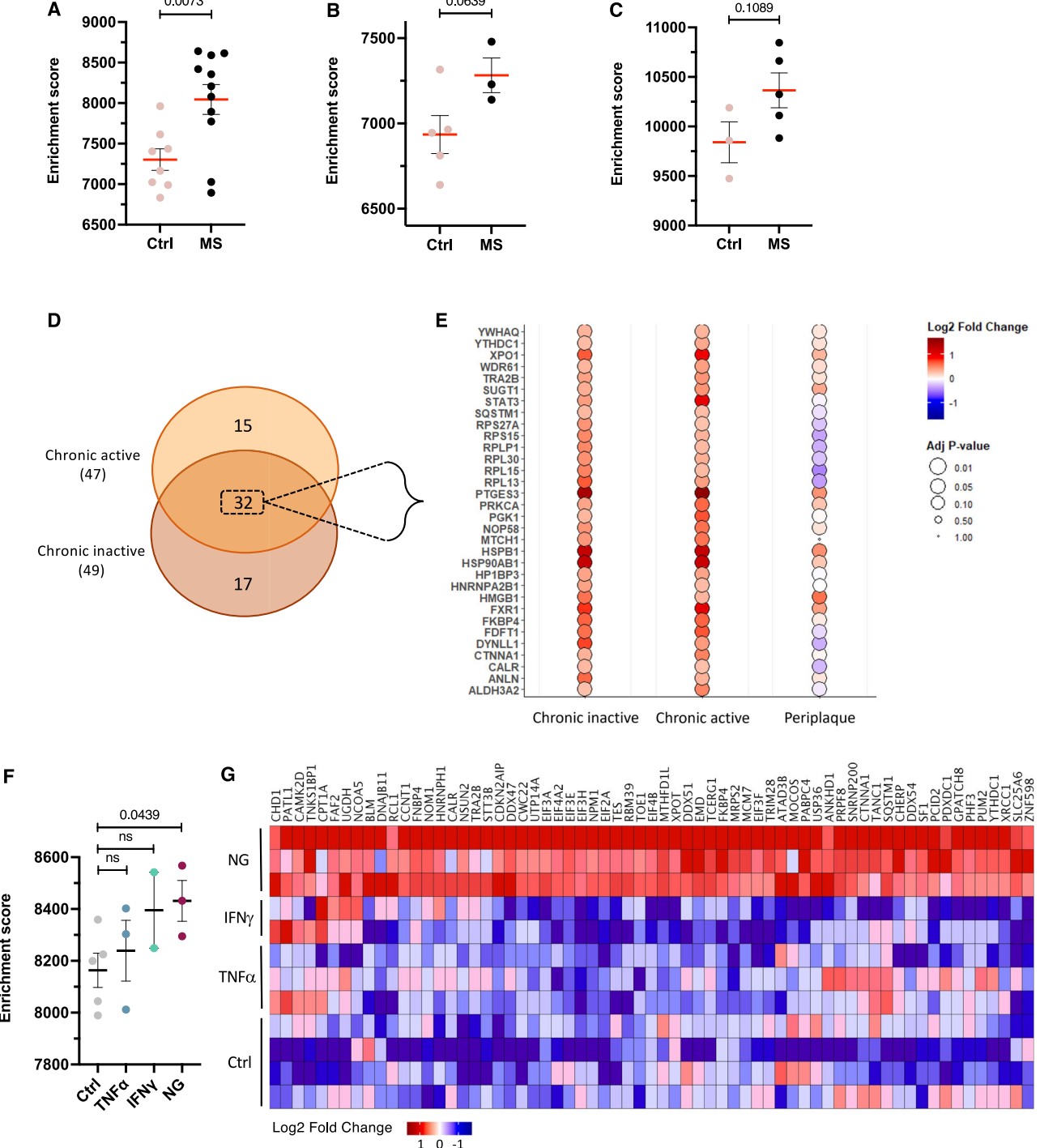

**Fig. 7 | ScRNA sequencing analysis of SG-associated genes in OL population in MS patients.** These analyses were performed on OL cell population exclusively using the SG-associated gene list recently established by the Wang group (26). **A–C** Molecular signature of OL cell population from publicly available datasets initially generated by the Schirmer group, (12) the Jäkel group, (25) and the Absinta group. (11) Single sample gene set enrichment analysis (ssGSEA) showing OLs from Schirmer (**A**) dataset (Ctrl, $n = 8$; MS, $n = 11$), Jäkel (**B**) dataset (Ctrl, $n = 5$; MS, $n = 3$) and Absinta (**C**) dataset (Ctrl, $n = 3$; MS, $n = 5$). **D** Euler Venn diagram of the significantly upregulated genes in MS patients compared to controls. The analysis was performed on the chronic active and chronic inactive lesions of the Absinta dataset. **E** Bubble plot of logFC depicting the 32 shared SG-related genes between the chronic active and chronic inactive lesions from (**D**). The color scale indicates logFC expression as compared to control groups; the size of the bubble denotes adjusted

$p$ value of genes expressed in MS compared to control groups. The periplaque area is added here as a comparator. **F**, **G** Bulk RNA sequencing analysis of SG-related genes in hOLs after 2 days of treatment of TNFα ($n = 3$), IFNγ ($n = 2$) or NG ($n = 3$) conditions using the list established by the Wang group. (26) The ssGSEA analysis compares the molecular signature of hOLs exposed to the corresponding treatments (**F**). The heat map shows the logFC of significant upregulated genes in NG conditions amongst the different treatments (**G**). Columns indicate individual samples grouped by treatment; the color scale indicates logFC expression as compared to control groups. Each dot in the graphs corresponds to an independent biological replicate. All data are expressed as mean values ± SEM, analyzed by unpaired two-tailed Student's $t$ test. All significant $P$ values are indicated; ns or unlabeled not significant. Source data are provided as a Source Data file.

## Molecular analysis indicates an increased expression of SG-related genes under in vitro metabolic stress conditions

We used our previously reported bulk RNA sequencing dataset, derived from in vitro experiments with hOLs exposed to TNFα, IFNγ, or NG conditions for 2 days, to examine the expression of the SG-associated molecules described by Wang et al [28]. At this time, sublethal injury was present in the OL population in all treatments[16]. The ssGSEA results showed a significant upregulation of a large number of SG-associated genes in hOLs exposed to NG conditions compared to controls (Fig. 7F). TNFα or IFNγ did not induce significant differences. The heatmap, present in Fig. 7G shows the logFC of the significantly upregulated genes under NG conditions as compared to control conditions. The associated downregulated genes are provided in Fig. S5B. Known functions linked to these genes include mRNA metabolism such as mRNA splicing and transport from the nucleus (PCID2, XPOT, and YTHDC1)[45–48] and also to cytoplasmic regulatory processes and RNP granule assembly (hnRNPH1, PABPC4, and PATL1)[49–51] (Fig. 7G). Our molecular analysis supports our in vitro immunostaining evidence and indicates that chronic metabolic stress conditions induce the upregulation of SG-related gene biology.

## Discussion

Our in situ analyses, derived from long-established cases with extensive disease pathology, show expression of SGs in OLs in active areas of MS lesions even in regions without extensive OL cell loss, persistence in inactive lesion areas where OL numbers are consistently reduced, and in NAWM. Although not the primary focus of our study, we additionally reported SG formation in some astrocytes in MS samples (Fig. S2A).

Our analysis of the ISR and mTOR protein translation pathways in these samples showed that the ISR marker ATF4 is highly expressed in OLs in the active areas of MS lesions, whereas there is reduced expression of the mTOR pathway marker p4E-BP1. These pathways act cooperatively to control global protein synthesis[52]. ATF4, the master regulatory protein of the ISR pathway, mediates functional interplay between these two signaling pathways by increasing the expression of 4E-BP1[53]. The phosphorylation of 4E-BP1 leads to its dissociation from the translation initiation factor eIF4E, allowing the formation of the initiation complex at the 5′-cap of mRNAs resulting in protein translation[54,55]. The observed prominent decrease of both ATF4 and p4E-BP1 in chronic inactive regions, together with the persistence of SGs even as OLs are lost in these areas, likely reflects the more severe loss of cell energy needed for protein synthesis and SG clearance. In NAWM, changes in ATF4 and p4E-BP1 expression are less than in the active areas but values still differ from control tissues, suggesting ongoing but as yet non-lethal injury.

Our access to surgical tissue samples from which we obtain viable OLs suitable for tissue culture has allowed us to assess the dynamics of SG formation and persistence in these cells under acute (SA) and chronic stress (NG) conditions. In comparison to the rapid formation and persistence of SGs in response to SA, SG formation was slower in response to NG conditions and was transient. The addition of pro-inflammatory cytokines IFNγ and TNFα to metabolic stress conditions resulted in SG persistence although these mediators themselves did not induce SG formation. Interestingly, the latter observation contrasts with previous studies showing formation of SGs under those immune factors in primary mouse neurons[56]. Our results suggest an impact on SG disassembly processes, which are also energy-dependent. These findings of the additive effect of IFNγ and TNFα with chronic stress help explain our in situ findings of persistent SG expression in OLs in MS lesions.

Although both the ISR and mTOR pathways and their interplay are implicated in promoting SG assembly and persistence under acute and chronic stress conditions[29,30], emerging evidence suggests differences in underlying mechanisms between the acute and chronic conditions[28,34,57]. In our system, SA induced a greater reduction in ATP and protein synthesis and increased cell death compared to NG conditions. As shown, OLs treated with NG conditions for 24 h displayed sustained translation inhibition, while no SGs were detected at this time. These observations are consistent with reported studies indicating that the mechanisms underlying SG persistence and those underlying inhibition of protein synthesis can be independent[58]."

Consistent with previous reports using cell lines, we found that SG formation in hOLs under acute stress conditions is dependent on the ISR pathway, being blocked by ISRIB. However, boosting the ISR with its agonist Sephin1 was not sufficient to induce SGs. It should be noted that our previous studies indicated that IFNγ and TNFα while causing sublethal injury to the OLs, did not induce the ISR pathway[16]. SG formation under chronic stress conditions is reported to be dependent on the ISR (eIF2α phosphorylation) and not on decreased mTORC1 activity[34]. The transient formation of SGs we observe under our NG chronic stress conditions is not blocked by ISRIB although we find activation of the ISR. Inhibiting mTORC1 activity, using the Torin1 inhibitor, resulted in increased level of SG formation and their subsequent persistence under both NG and control conditions. These findings implicate stress-associated reduction in mTOR pathway activity as a major determinant of SG assembly and disassembly in hOLs.

Our studies using glucose/glycogen inhibitors indicate that the distinct glycolysis-dependent metabolic properties of hOLs contribute to the formation and persistence of SGs. The transient SG formation induced by NG conditions became persistent when glycolysis was blocked by either 2DG or CP agents. We postulate that the glycolytic features of hOLs allow them to respond to metabolic stress conditions by using glycogen as an emergency energy source. This adaptation may be responsible for the initial SG disassembly in hOLs in vitro under NG conditions. These observations are consistent with previous reports on iPSC-derived cortical organoids from ALS patients, showing that the glycolysis pathway regulates the balance between the formation and clearance of SGs by maintaining the intracellular ATP pool[28]. The impact of reduced energy and perturbed glycolytic metabolism would be consistent with data from imaging studies showing hypometabolism in MS patients[4,5,33].

Although our focus is on how molecular modifications associated with SG formation occur post-translationally, here we have also used three different publicly available scRNA-seq datasets to document the transcriptomic adaptation of stressed OL populations derived from human MS brain samples. We show that changes in SG-related genes are linked with chaperones and autophagy processes, known to impact SG biology. We consider that a prolonged stress environment and chronic SG formation could induce a long-term modification and molecular adaptation at the transcriptional level to modulate the processes of SG assembly or disassembly.

Still unresolved is the function SGs serve or what advantages stress-induced condensation imparts during the injury response of the cell. Although the response of the ISR and mTORC1 pathways to stress conditions are primarily thought to be cytoprotective[59,60], sustained inhibition of protein synthesis may become particularly deleterious to hOLs, which must be able to maintain the turnover rate of their extensive myelin processes[61,62]. Reineke and colleagues showed that removal of SGs reduced cell death in response to chronic nutrient starvation, raising the possibility that, in contrast to acute stress-induced granules that support cellular survival, some conditions such as glucose deprivation induce pro-death SGs[34]. Under our metabolic stress conditions, SG appeared and disappeared prior to cell death becoming apparent. Whether this transient SG formation reflects an attempt at neuroprotection remains speculative. Our study characterizes features of glucose deprivation induced-SGs that are distinct from canonical SGs, as suggested before[63].

Mechanistically, SGs reflect the spatiotemporal regulation of mRNA translation and protein synthesis and are deemed a "decision point" for the fate of mRNA and proteins trapped in their structure[30,42]. While transient SG formation is considered a protective response for

cells to overcome stress, persistent granules are implicated in the pathogenesis of neurodegenerative diseases, including MS[19,64,65]. Perturbation of SG dynamics is proposed to precipitate the toxicity of disease-related proteins such as hnRNP A1 or TDP-43, eventually leading to protein aggregations and cellular death, although this remains controversial[66]. Our in vitro data confirmed our previous observation[66] that some RBPs become mislocalized from the nucleus under metabolic stress conditions. We now show that PABP and to some extent, hnRNP A1 coalesce with G3BP1-labeled SGs, which is not the case for TDP-43. Our data support that SG composition is not static and changes over time, as previously mentioned[63,67].

We suggest that persistence of SGs in OLs in MS reflects distinct changes in the protein translation regulatory pathways and glycolytic metabolic properties of these cells in response to a combination of metabolic stress and pro-inflammatory conditions.

## Methods

The use of human rapid post-mortem brain tissue samples was obtained with full ethical approval (protocols BH07.001.31; Nagano 20.332-YP) and informed consent from the Neuroimmunology Research Laboratory, Centre de Recherche du Centre Hospitalier de l'Université de Montréal (CRCHUM). The use of anonymized adult surgical samples was approved by the Montreal Neurological Institute Research Ethics Board (protocol ANTJ 1988/3) and the use of pediatric surgical samples by McGill University Health Center Research Ethics Board (protocol 13-244-PED). Written informed consent was obtained from responsible guardians for every pediatric donor prior to sample collection.

### Antibodies and reagents

Immunohistochemistry−Sudan Black solution was purchased from Millipore Sigma (#S2380). Anti-Nogo-A (11C7) antibody was provided by the Brain Research Institute, University of Zurich (17356385). G3BP1 (#181150), PABP (#21060), ATF4 (#31390), and anti-GFAP (#ab4674) antibodies were purchased from Abcam. p4E-BP1 (#2855S) antibody from Cell Signaling.

Immunocytochemistry−Anti-O4 antibody (#MAB1326) was purchased from RnDsystems. Anti-mouse (#56574) or anti-rabbit (#181150) G3BP1, PABP (#21060), and hnRNP A1 (#4791) antibodies were purchased from Abcam. Phosphorylated eIF2α (#701268) and phosphorylated 4E-BP1 (#2855S) from ThermoFisher and Cell Signaling, respectively. TDP-43 (#10782-2-AP) antibody was purchased from Proteintech.

Secondary antibodies−Goat anti-rabbit (#A11034) and anti-mouse (#A21121) Alexa Fluor 488; goat anti-rabbit (#A21428) Alexa Fluor 555 and Hoescht (#33258) were purchased from Invitrogen. Alexa Fluor 647 (#1021-31) from Southern Biotech and goat anti-rabbit CY3 (#111-166-047) from Jackson ImmunoResearch.

Reagents−TNFα (PHC3016) and IFNγ (PHC4031) were purchased from ThermoFisher Scientific. Sephin1 was purchased from ApexBio (#A8708). ISRIB (#SML0843) and Cycloheximide (#4859) from Sigma. Torin1 (#4247) from Tocris Bioscience. 2-deoxy-D-glucose (2DG) and Sodium Arsenite from Sigma. CP (91149) from Selleckchem.

### In situ immunohistochemistry

Human rapid post-mortem brain tissue samples were obtained with full ethical approval and informed consent from the Neuroimmunology Research Laboratory, Centre de Recherche du Centre Hospitalier de l'Université de Montréal (CRCHUM)[68]. Areas of ongoing demyelination were selected using LFB and H&E staining based on the presence of macrophages, some of which contained LFB-positive material (Fig. S1A). There were also areas of inactivity in these active/mixed lesions. Patients characteristics are showed in the supplementary material (Table S1). Immunohistochemistry procedure was performed using immunofluorescence[69]. Primary antibodies used were mouse anti-Nogo-A (1:5000), rabbit anti-ATF4 (1:200), rabbit anti-p4E-BP1 (phosphorylated in Thr37/46, 1:1600) and Hoescht (1:1000). Secondary antibodies used were goat anti-rabbit Alexa Fluor 488 (1:500) and goat anti-mouse Alexa Fluor 555 (1:500). Sudan Black solution (0.3%) was applied for 3 min and rinsed extensively with PBS before the sections were mounted on coverslips. Sudan Black was used to remove endogenous autofluorescence due to lipofuscin. As controls for non-specific binding, primary antibodies were omitted. Images (z-stacks) were acquired using a Leica TSC SP8 confocal microscope and processed using ImageJ or Imaris. Microscope settings (image exposure times and thresholds) were kept the same during the image acquisition of all the biological samples. Number of cells (NogoA+) per mm$^2$ was compared between active and inactive areas and NAWM of MS patients and WM from the control tissues. Marker expression intensity quantification was performed using the following scoring system[16]: no expression (absence of signal), low-intensity signal, or high-intensity signal. For SG quantification, a cell was scored as positive when it had at least three foci. Data were derived by blinded observers counting >150 cells per selected area when possible.

In the present study, analyses were done on two controls and 3 MS patients accounting for independent regions as 2 WM, 4 NAWM, 12 active and 3 inactive areas.

### Human cell samples

Human brain tissue samples resected during surgical procedures were processed for cell isolation within 1–2 h. Tissue samples were collected into CUSA bags from the surgical "corridor" away from the site of pathology. The material comprised predominantly white matter tissue and focal or diffused pathologic was excluded as possible during the operative procedure and retrospective histological analyses. The clinical details of human brain tissue samples used for in vitro investigations are provided in the supplementary material (Table S2). The use of adult surgical samples was approved by the Montreal Neurological Institute Research Ethics Board and the use of pediatric surgical samples by the Montreal Children's Hospital Research Ethics Board. Single cells were obtained from human brain tissue samples after trypsin digestion followed by Percoll gradient centrifugation[70]. To obtain an enriched population of mature OLs, the dissociated cells were cultured overnight in Dulbecco Modified Eagle Medium-F12 (DMEM/F12) medium supplemented with N1 (Sigma), 0.01% bovine serum albumin, and 1% penicillin-streptomycin (Invitrogen) in noncoated flasks. The floating cell fraction comprised primarily of mature OLs was then collected for use in experiments, leaving behind adherent microglia. The purity of human OL cell culture in vitro is regularly checked by immunofluorescence using O4 antibody and estimated at >95% purity.

### Human OL cell culture and treatment

After the isolation procedure, primary human OLs were plated into 12 well poly-L-lysine and extracellular matrix-coated chamber slides (30,000 cells per well) in optimal medium consisting of DMEM-F12 supplemented with N1, 1% P/S, B27 (Invitrogen), platelet-derived growth factor with 2A subunits (10 ng/mL), basic fibroblast growth factor (10 ng/mL), and tri-iodothyronine (2 nM) (Sigma).

Cultures were maintained for an initial week under optimal conditions before subjecting to experimental conditions, to allow cell adherence and process formation. Effects of the experimental conditions in dissociated cultures were assessed for up to 4 days. For metabolic experiments, cells were cultured in a less enriched medium (DMEM-F12 without N1 and supplementary factors) containing 0.25 g/L of glucose or with no glucose added. We refer to these conditions as Low Glucose (LG) and No Glucose (NG), respectively. Mediators TNFα (100 ng/mL) and IFNγ (2000 IU per mL) were diluted in the optimal medium. Sephin1 (5 µM) and ISRIB (200 nM) treatments (defined from ref. 16) were conducted under Ctrl and stress conditions. Both drugs were added at the onset of the treatment for 2, 4, 8, and 24 h. Torin1

(1 μM, defined from[71]), was used in Ctrl and stress conditions for 4, 8, and 24 h. The glycolytic pathway inhibitors, 2-DG (2.5 mM) and CP (50 μM) were used in optimal and stress conditions for 4, 8, and 24 h. Working concentrations were based on the range of previous study[28].

### Induction of stress granules in vitro

To induce SGs, hOLs were treated with 0.5 mM SA for the indicated times at 37 °C. Recovery was initiated by replacing the stress media with the optimal culture medium.

### In vitro immunocytochemistry

The staining procedure was conducted as described before[16]. Briefly, hOLs were incubated with a monoclonal mouse primary O4 antibody for 15 min at 37 °C and then fixed in 4% paraformaldehyde for 10 min at room temperature. Cultures were permeabilized with 0.3% Triton X-100 in PBS followed by blocking with HHG (1 mM hydroxyethylpiperazine ethanesulfonic acid, 2% horse serum, 10% goat serum, Hank's balanced salt solution). Cells were incubated with the following primary antibodies for 1 h at room temperature: SG markers −rabbit anti-G3BP1 (1:300) or mouse anti-G3BP1 (1:500, Abcam). SG composition−rabbit anti-PABP (1:500), rabbit anti-hnRNPA1 (1:1000) and rabbit anti-TDP-43 (1:500). ISR and mTOR pathways−rabbit anti-peIF2a (phosphorylated in Ser51, 1:400) and anti-p4E-BP1 (phosphorylated in THr37/46, 1:200). After being washed with PBS, the cells were incubated with goat anti-mouse or anti-rabbit Alexa Fluor 488 (1:500, Invitrogen), Alexa Fluor 555 (1:800) or Alexa Fluor 647 (1:500), together with nuclei staining with DAPI (Hoechst, 1:1000) for 1 h. As controls for non-specific binding, primary antibodies were omitted. Images (z-stacks) were acquired using a Leica TSC SP8 confocal microscope and processed using ImageJ or Imaris. Microscope settings (image exposure times and thresholds) were kept the same during the image acquisition of the related intensity expression experiments. Semi-quantitative analysis of stress markers expression (p-eIF2α and p4E-BP1) was carried out using ImageJ and the following scoring system: no expression (absence of signal), low-intensity signal or high-intensity signal. For SG quantification, a cell was scored as positive when it had at least three foci. Data were derived by blinded observers counting >100 cells per time point, per condition, and experiment.

### Cell viability

hOLs viability was assessed by live staining using propidium iodide dye (PI; ThermoFisher Scientific). The average number of PI+ (dead cells) and total cell number (Hoechst) per condition was determined via Leica TSC SP8 confocal microscope and ImageJ. For apoptosis assay, cells were incubated in the presence of Cell Event Caspase 3/7 detection reagent (C10423, Invitrogen) for 1 h at 37 °C.

### ATP analysis

ATP levels were detected with the luciferase-based Cell Titer-Glo 2.0 (Promega, G9242) following the manufacturer's protocol. Briefly, cells were seeded into 96-well plates and cultured under optimal, SA or NG conditions for the indicated time. The Cell Titer reagent was mixed with cell culture medium and incubated for 10 min and measured on a luminescent plate reader. After treatments, the cell number per well was determined using Hoechst staining (1:1000) and finally normalized with luciferase values.

### Protein synthesis assay

Detection of nascent protein synthesis was assessed with Click-IT protein synthesis assay kit (#C10428, ThermoFisher) and performed following the manufacturer's protocol. The methionine analog L-homopropargylglycine was added (50 μM) to cultures 30 min before the first sampling of each time point. The detection of click reaction was obtained by incubating fixed cells for 1 h with Alexa Fluor 488 azide in Click-iT cell reaction buffer solution. Fluorescence was recorded using a Leica TSC SP8 confocal microscope and ImageJ. Nascent protein synthesis was assessed by determining the signal intensity and finally normalized by the control condition of each biological replicate. Microscope settings (image exposure times and thresholds) were kept the same during the image acquisition of all the different conditions.

### RNA sequencing

Bulk sequencing−We utilized our published datasets on hOLs[16]. RNA was extracted from hOLs after 2 days in culture under control (optimal), NG, TNFα and IFNγ conditions. The clinical details of human brain tissue samples used for in vitro molecular investigations are provided in the supplementary material (Table S3). RNAseq was performed at the Génome Québec Centre using the Illumina platform and NovaSeq 6000 PE100 machine. We used the GenPipes workflow for alignment to the GRCh38 human genome and read counting. Raw FastQ files were aligned to the GRCh38 genome reference using the STAR aligner with default parameters[72]. Raw reads were quantified using HTSeq count. Raw read counts were then normalized and variance-stabilized. DESeq2 package in R was employed for further analysis[73]. Significantly differentially expressed genes were filtered using a p value cutoff of <0.05. Heatmaps were generated after clustering using the Hierarchical Clustering Image function in GenePattern (v3.9.11). Gene clustering was performed using Pearson correlation and the normalized values were transformed using a long transformation. Finally, row normalization was applied for the final visualization[74].

Single nuclear RNA sequencing−The snRNAseq data of human MS and control patients were obtained from ref. 10; 11 MS patients, 9 controls), ref. 39; 4 MS patients, 4 controls), and ref. 9; 5 MS patients, 3 controls) datasets. The datasets were subjected to the standard Seurat pipeline for dimensionality reduction, cell type identification, gene expression normalization, and clustering[75]. OL clusters were chosen in each dataset using canonical OL markers and information provided by the reference publications. Average expression and differentially expressed genes (DEG) between MS and control OLs were identified using a log2(fold change) > 0.25 and an adjusted p value < 0.05 as cutoffs. The log2FC and adjusted p values were utilized to generate a bubble plot using ggplot2 v3.3.6 in R. A Venn diagram was employed to visualize the overlap between significant genes in the edge of chronic active and the edge of chronic inactive lesions using the Venny 2.1 web tool.

### Statistical analysis

In RNA-seq analyses, the GenePattern platform[74] was utilized for single-sample gene set enrichment analysis (ssGSEA) to provide pathway-level enrichment in both bulk RNA-seq and snRNA-seq datasets. We employed the protein list established by Wang et al.[28] to create a signature for the ssGSEA analysis (Data file S2). For the bulk RNA-seq datasets, we utilized the normalized read counts of the control (optimal), NG, IFNγ, and TNFα samples for ssGSEA analysis. As for the snRNA-seq datasets, we used the average expression of OL markers for both patient and control samples from the Schirmer, Jäkel, and Absinta studies. The resulting enrichment scores (ES) were used to generate graphs in Prism 9.1−GraphPad, and p values were calculated using Student's t test for comparisons between groups.

In cell culture analyses, we performed each experiment with at least three independent biological replicates, unless otherwise specified in the figure legend. In all of the imaging analyses, an observer who was blinded to the experimental groups conducted the quantification. The data were analyzed using GraphPad Prism 9.5 software and are presented as means ± SEM. Student's t test was used for comparisons between two groups. Comparison between three or more independent groups was performed using one-way ANOVA, followed by Bonferroni's multiple comparison correction. P values < 0.05 were considered statistically significant.

**Reporting summary**

Further information on research design is available in the Nature Portfolio Reporting Summary linked to this article.

## Data availability

All data generated or analyzed during this study are included in this article and its supplementary information files. Source data are provided with this paper. Bulk sequencing data from this study have been deposited into the Gene Expression Omnibus (GEO) database (GSE249381). Publicly available single nuclear RNA sequencing data are available under: NCBI Bioproject 544731; Accession number GSE118257; Accession number GSE180759. Source data are provided with this paper.

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

## Acknowledgements

Images were collected and/or image processing and analysis for this manuscript was performed in the McGill University Advanced BioImaging Facility (ABIF). This work was supported by a grant (BRAVE in MS) from the International Progressive MS Alliance (PA-1604-08492) and Le Grand Portage Foundation (J.P.A.). A.P. holds the T1 Canada Research Chair in Multiple Sclerosis and is funded by the Canada Institute of Health Research, the National Multiple Sclerosis Society, and the Foundation of Canada for Innovation. C.V.V. is funded by the Canada Institute of Health Research, ALS Canada/Brain Canada, and the Natural Sciences and Engineering Research Council.

## Author contributions

Conceptualization: FP, QLC, and JPA. Methodology: FP, QLC, MGFF, HES, MCL, MS, RD, SEJZ, WK, AP, GRWM, CVV, JPA. Investigation: FP, QLC, HES, GRWM. Resources: JF, MS, RD, SEJZ, WK, AP, HES, MCL. Formal analysis: FP, QLC, AM, MGFF. Writing: FP, AM, GRWM, TEK, CVV, JPA with editing and discussion from all coauthors. Funding acquisition and supervision: JPA.

## Competing interests

The authors declare no competing interests.
