## [Peer Review File · Nature Communications]

REGULATION OF STRESS GRANULES FORMATION IN OLIGODENDROCYTES AS APPLIED TO MULTIPLE SCLEROSISREVIEWER COMMENTS

Reviewer #1 (Remarks to the Author):

The manuscript from Pernin and colleagues examines stress granule formation in human oligodendrocytes in vivo and in vitro. The authors detected stress granules in oligodendrocytes in autopsy samples from MS patients, both in the lesions as well as in normal appearing white matter. In addition, the authors explored the dynamics of stress granule formation in oligodendrocytes isolated from primary mature oligodendrocytes isolated from surgically resected brain tissue. The authors found that stress granules could be transiently induced in these cells when cultured in the absence of glucose but not by the presence of inflammatory cytokines, although when combined with the metabolic stress, the stress granules persisted. Their in vitro data suggests that oligodendrocytes in MS patients are subjected to metabolic and inflammatory stress, which explains the widespread presence of stress granules in vivo. The study is well done and nicely combines the immunohistochemical analysis of human autopsy tissue with mechanistic studies in culture. Overall, the study is well done, although the in vivo significance of the cell culture work needs to be enhanced. Consideration of the following points should strengthen the manuscript, by increasing the implications of the study for the in vivo environment of MS lesions.

1. The authors need to better support the relevance of their "metabolic stress" culture conditions. The absence of glucose appears excessively harsh, especially since the low glucose media did not result in a similar impact on the cultured oligodendrocytes. How does the absence of glucose culture media model the in vivo environment of the MS CNS?
2. The criteria used to distinguish active versus inactive MS lesions needs to be described in the results section.
3. Some descriptions of the data are baffling. On the top of page seven they describe 49% of oligodendrocytes expressing ATF4 as "high", yet in the next sentence they describe 41% of oligodendrocytes expressing p4E-BP1 as an indication of repressed expression. The description of this data needs clarification.
4. The authors examine IFN-gamma and TNF-alpha separately and show that neither can induce stress granules in cultured human oligodendrocytes. They combine these cytokines with media that lacks glucose and show that together stress granules persist. It would be interesting to determine if the combination of IFN-gamma and TNF-alpha, in normal glucose levels, impacts stress granule formation. Otherwise, the conclusions from this experiment are overstated. It would be also interesting to combine the inflammatory cytokines with the low glucose media to determine if there is an added effect on stress granule formation.
5. It is not clear what the significance of the ATP results are. Have the authors examined the impact of inflammatory cytokines on oligodendrocyte ATP levels? It is not particularly informative that the absence of glucose results in reduced ATP levels.
6. Similarly, the authors should examine the impact of inflammatory cytokines on protein synthesis in their human oligodendrocyte cultures, alone and in combination with reduced levels of glucose, to determine if there is synergy. This would provide a clearer link to the in vivo environment present in MS patients.
7. The same argument, as described in points 4, 5, and 6, can be made with regard to the ISRIB, Sephin1 and Torin1 inhibitor studies. By combining with the inflammatory cytokines, the results would be more relevant to the MS CNS environment.

Minor Point: some of the headings of the results section appear to be subheadings, which creates a bit of confusion.

Reviewer #2 (Remarks to the Author):

This manuscript is dedicated to study how oligodendrocyte injury and subsequent loss, which is a pathologic hallmark of multiple sclerosis (MS), may be linked to stress granule (SGs) biology. While authors have presented some interesting findings, the conclusions drawn here are not well supported by their data. To be considered for the publication, this work should be significantly strengthened by additional data.

The main issue with this work is that data presented is quite descriptive and do not provide any detailed mechanisms. Detection of SGs in OLs in active and inactive areas of MS lesions as well as in normal-appearing white matter is not surprising as well as induction of transient SGs in cultures of primary human adult brain derived OLs in response to metabolic stress. Persistent SG formation in response of combining pro-inflammatory cytokines and metabolic stress is an interesting phenomenon, which is not explained mechanistically. Why are they persistent? What do pro-inflammatory cytokines do with canonical SGs? Is there evidences that SG persistence is the result of failure of SGs to disassemble?

Specific issues:

1. Authors should use more markers for SGs to demonstrate co-staining of PABP+ foci with other SG components. Do polyadenylated mRNAs present in these foci?
2. Puromycin/ cycloheximide treatments should be used to demonstrate that SG-like cytoplasmic foci are actually dynamic
3. How pharmacological interventions (puromycin/ cycloheximide) with protein synthesis affect "persistent SGs"?
4. What is actual significance of PABP and hnRNPA1 coalescence with G3BP1-labeled SGs? Or the absence of TDP-43? What this mislocalization findings tell us?
5. Do authors think that main observations of this work obtained in OLs are unique to OLs or it is common phenomenon for other cell types
6. Finally, it is important to know what is actually happening in diseased state (MS) since majority of the experiments are done using human primary mature OLs.

Reviewer #3 (Remarks to the Author):

The manuscript by Pernin investigates the role of stress granules in oligodendrocytes focusing on the pathological condition in multiple sclerosis (MS). Stress granules is well documented in many pathological conditions, but their functions, regulations, and therapeutic implications are under studied. Moreover, their presence in MS has never been intensively studied, therefore this manuscript has great importance to the field. The investigators first confirm using human multiple sclerosis brain samples the presence of stress granules. Comparison to the control brain tissues has provided clear evidence for the presence of stress granules in MS. The investigators then used in vitro model to demonstrate types of stressors that initiate the formation of stress granules in human oligodendrocytes. The use of human oligodendrocytes is another important piece of validation for stress granules. Lastly, the investigators confirm their finding using publicly available RNA-seq data from 3 independent studies. These analyses confirm the presence of stress granule related genes in MS tissues. Taken together, the investigators have gathered convincing evidence

for the presence of stress granules in MS, which provides substantial advance to the field. The quality of each experiment is high, demonstrating replicability across publicly available datasets. Despite above mentioned strengths of the study, lack of critical information makes it difficult to assess the validity of some aspects of experiments. Investigators are requested to provide further information explained below, and also make modifications to the figures suggested below.

1. Abstract – the word “ISIRB” need to be defined.
2. Fig 1 – the authors state that “Lesion activity was characterized by the presence of macrophages/microglia throughout the lesion area” to define MS regions. An independent figure is needed to demonstrate this assessment. Clarification is needed to show how investigators delineated NAWM, active, inactive areas.
3. Fig 1 A/B – Nogo immunofluorescence in control WM seems lower in panel B than the actual quantification shown in panel A. Is this truly a representative image?
4. Fig 1B/C – inactive area does not show PABP staining colocalizing with Nogo, in contrast to the quantification in panel C. Is this truly a representative image?
5. Fig S1C – the demonstration of DAB staining does not show colocalization with oligodendrocyte marker(s) therefore does not add further validity to what is already stated in Fig1. It is recommended to remove this panel.
6. Fig 1D – needs to increase the contrast for ATF4 fluorescence. It shows very dim, making it difficult to assess the claim.
7. Fig 1F - needs to increase the contrast for p4E-BP1 fluorescence. It shows very dim, making it difficult to assess the claim.
8. Fig 2 and Fig 4 – the magenta color needs to be labeled as “O4”, and blue needs to be labeled “Dapi” within the panels, just like G3BP1 is labeled within the panels.
9. Fig 6C – it is recommended to increase the contrast of TDP-43. The fluorescent is dim and difficult to assess the claim.
10. Although Fig. 2 legend states N is ‘independent biological replicate’, additional explanation needs to be included regarding how quantification was made, such as the number of total cells/fields of interest/technical replicates quantified. This can be in the method section or in the legend.
11. Study design – There needs to be a comprehensive table with description of MS patient and control tissues including number of cases, age, type of MS, etc. Also needed is the explanation of type of surgically resected tissues (quantity, pathology, age, criteria for use in case of pathological tissues, etc). Throughout the manuscript, lack of these information makes it difficult to assess the replicability of findings.
12. Human cell samples – the authors state “human OL cells culture in vitro is regularly checked by immunofluorescence using O4 antibody and estimated at >95% purity”. This needs to be demonstrated in an independent figure. Without the validation of OL culture purity and viability, it is difficult to assess the validity of in vitro experiments.

Reviewer #4 (Remarks to the Author):

Identifying the mechanisms mediating oligodendrocyte (OL) injury at distinct stages of MS represents a critical topic in the field with major therapeutic implications. Substantial prior evidence has suggested that both metabolic and classic inflammatory pathways likely contribute, particularly in chronic stages of MS. The work presented here addresses this question in an insightful manner, focusing on the relationship of stress granule (SG) formation to metabolic and inflammatory stressors. The authors convincingly demonstrate that SG formation occurs in OLs of MS patients, including NAWM and lesions of distinct pathologic stages, and they relate these findings to disruption of protein translation, energy deficiency, and cell death. They nicely demonstrate the distinct and yet synergistic effects of glucose deprivation and inflammatory cytokine stimulation, altogether providing important new insights into potential drivers of OL injury in MS. This is an important and relevant study. I have only a few minor comments to address:

-At the beginning of the results section, a brief description of the human MS and control tissue sample set would be helpful to understanding the data. Some details are provided in the Methods, but the readers’ understanding would be improved if these were included in the main text. If possible, a simple supplementary table with these details would also be helpful. Important details

include how many human subjects were included, age, sex, and disease duration (if known).

-In Figure 1, it seems surprising that the number of NogoA+ cells was only mildly decreased in active lesions compared to control WM and MS NAWM. This raises a couple questions. First, can more detail be given about how lesions were identified and classified? This was mentioned only briefly. Was classification consistent with prior pathologic studies in human MS post-mortem samples? Second, although NogoA is used as a marker of OLs, it is not exclusively expressed in OLs and has been reported in neurons among other cell types. Were NogoA+ cells clearly identified as OLs in this study? And how was this accomplished?

-With no glucose (NG), protein translation remains low at 24 hours despite SG formation being only transient (resolved by about 8 hours). This would suggest that SG formation is not the only determinant of translation in OLs. Can this be addressed in the text or discussion?

Reviewer #1 (Remarks to the Author):

The manuscript from Pernin and colleagues examines stress granule formation in human oligodendrocytes *in vivo* and *in vitro*. The authors detected stress granules in oligodendrocytes in autopsy samples from MS patients, both in the lesions as well as in normal appearing white matter. In addition, the authors explored the dynamics of stress granule formation in oligodendrocytes isolated from primary mature oligodendrocytes isolated from surgically resected brain tissue. The authors found that stress granules could be transiently induced in these cells when cultured in the absence of glucose but not by the presence of inflammatory cytokines, although when combined with the metabolic stress, the stress granules persisted. Their *in vitro* data suggests that oligodendrocytes in MS patients are subjected to metabolic and inflammatory stress, which explains the widespread presence of stress granules *in vivo*. The study is well done and nicely combines the immunohistochemical analysis of human autopsy tissue with mechanistic studies in culture. Overall, the study is well done, although the *in vivo* significance of the cell culture work needs to be enhanced. Consideration of the following points should strengthen the manuscript, by increasing the implications of the study for the *in vivo* environment of MS lesions.

Response - We appreciate the positive comments about the study being well done and combining *in situ* and tissue culture analyses. Below we address the specific issues raised to enhance the implications of the *in vitro* findings for the MS lesion environment.

Major points:

1. The authors need to better support the relevance of their “metabolic stress” culture conditions. The absence of glucose appears excessively harsh, especially since the low glucose media did not result in a similar impact on the cultured oligodendrocytes. How does the absence of glucose culture media model the *in vivo* environment of the MS CNS?

Response - The low glucose (LG) conditions will cause increased cell death compared to optimal (Ctrl) conditions if the culture length is extended e.g., 6 days (Cui et al., 2017 – reference 67). The no glucose (NG) conditions provide a more rapid response.

We agree it is difficult to exactly match *in vivo* and *in vitro* conditions with regard to glucose levels as the optimal culture conditions require significantly higher levels of glucose compared to levels in the brain (Glucose level: Ctrl=17.5mM, LG conditions=1.4 mM, brain=2.5mM). Our “stress” culture conditions of low and no glucose (LG or NG) have reduced levels of other nutrients found in the enriched Ctrl conditions (N1 media).

As we describe in Pernin et al (Brain 2022 – reference 15), it is with the *in vitro* metabolic stress conditions that we can model the dying back phenomenon seen in OLs in MS lesions (reference 66). LG and NG conditions are the only stress conditions leading to cell death whereas pro-inflammatory cytokines or excitotoxins alone result only in process retraction.

We have revised our reference list to provide more definitive evidence of metabolic stress in MS lesions.

2. The criteria used to distinguish active versus inactive MS lesions needs to be described in the results section.

Response - As requested, we include our description of these two lesion categories (active vs inactive lesions) directly in the Results section: *“Lesion activity was characterized by LFB and H&E staining. Areas of ongoing demyelination were selected based on the presence of macrophages/microglia, some of which contained LFB-positive material (Fig. S1A). There were also areas of inactivity in these active/mixed lesions.”*

We provide a new supplementary figure of an active MS lesion (Fig.S1A) showing LFB-positive material in macrophages/microglia.

3. Some descriptions of the data are baffling. On the top of page seven they describe 49% of oligodendrocytes expressing ATF4 as “high”, yet in the next sentence they describe 41% of oligodendrocytes expressing p4E-BP1 as an indication of repressed expression. The description of this data needs clarification.

Response - Thanks to the reviewer’s comment, we made an error in the original manuscript which we have now corrected with regards to the “41%”. The corrected text reads as *“Only 19% ±13.4% of OLs in active lesion areas have high expression of phosphorylated 4E-BP1. In contrast, control patients demonstrate sustained mTOR activity and protein synthesis, as shown by a high expression of p4E-BP1 (85% ±3.5) (Fig. 1F & 1G).”*

The results section has been revised.

4. The authors examine IFN-gamma and TNF-alpha separately and show that neither can induce stress granules in cultured human oligodendrocytes. They combine these cytokines with media that lacks glucose and show that together stress granules persist. It would be interesting to determine if the combination of IFN-gamma and TNF-alpha, in normal glucose levels, impacts stress granule formation. Otherwise, the conclusions from this experiment are overstated. It would be also interesting to combine the inflammatory cytokines with the low glucose media to determine if there is an added effect on stress granule formation.

Response - We have now conducted the suggested experiments. We combined the pro-inflammatory cytokines (TNF α and IFN γ) with the normal glucose media (Ctrl) or with the low glucose media (LG). The cytokine combination did not alter SG formation under control or LG conditions.

These new data have been added to Fig. 2D-2F and described in the results section.

5. It is not clear what the significance of the ATP results are. Have the authors examined the impact of inflammatory cytokines on oligodendrocyte ATP levels? It is not particularly informative that the absence of glucose results in reduced ATP levels.

Response - As requested here and in item #6, we carried out ATP and protein synthesis experiments under the following conditions: Ctrl+TNF α +IFN γ and NG+TNF α +IFN γ

conditions. These new data are now provided in Figure 3A-C. We do not observe an effect of the cytokines on ATP or protein synthesis under control conditions and no synergistic effect under NG conditions.

6. Similarly, the authors should examine the impact of inflammatory cytokines on protein synthesis in their human oligodendrocyte cultures, alone and in combination with reduced levels of glucose, to determine if there is synergy. This would provide a clearer link to the in vivo environment present in MS patients.

Response - Please, see the response to comment #5, above.

7. The same argument, as described in points 4, 5, and 6, can be made with regard to the ISRIB, Sephin1 and Torin1 inhibitor studies. By combining with the inflammatory cytokines, the results would be more relevant to the MS CNS environment.

Response - As the reviewer suggested, we have conducted these experiments.

- With ISRIB, there was no reduction of SG formation at 24 hours under NG+ TNF α +IFN γ conditions, as seen with NG cultures alone. These new data are presented in Fig S3B.
- Adding Sephin1 to the Ctrl+TNF α +IFN γ cultures did not induce SG formation at 24h. The new data are presented in Fig S3A.
- Adding Torin1 to Ctrl+TNF α +IFN γ cultures resulted in similarly increased SG formation as seen with Ctrl cultures alone. These new data are presented in Fig S3A.

Minor point:

8. some of the headings of the results section appear to be subheadings, which creates a bit of confusion.

Response - We have revised the results section in line with this comment.

Reviewer #2 (Remarks to the Author):

This manuscript is dedicated to study how oligodendrocyte injury and subsequent loss, which is a pathologic hallmark of multiple sclerosis (MS), may be linked to stress granule (SGs) biology. While authors have presented some interesting findings, the conclusions drawn here are not well supported by their data. To be considered for the publication, this work should be significantly strengthened by additional data.

The main issue with this work is that data presented is quite descriptive and do not provide any detailed mechanisms. Detection of SGs in OLs in active and inactive areas of MS lesions as well as in normal-appearing white matter is not surprising as well as induction of transient SGs in cultures of primary human adult brain derived OLs in response to metabolic stress. Persistent SG formation in response of combining pro-inflammatory cytokines and metabolic stress is an interesting phenomenon, which is not explained mechanistically.

A. Why are they persistent?

B. What do pro-inflammatory cytokines do with canonical SGs? Check in the literature /context

C. Are there evidences that SG persistence is the result of failure of SGs to disassemble?

Response - In response to the above overview of our work, we submit that while detection of SGs in OLs in active and inactive areas of MS lesions as well as in normal-appearing white matter may have been predicted by some (“not surprising”), having the actual data is meaningful. We further submit that our *in vitro* studies do at least begin to provide insight as to how factors present in the MS lesion environment (metabolic stress, inflammatory mediators) impact on SG formation.

With regards to the term canonical SGs, we point out that the SGs we observed under the SA or NG conditions have distinct features. Canonical SGs induced under SA conditions are inhibited by the ISRIB molecule, suggesting that the major pathway involved in their formation is related to the phosphorylation of eIF2 α . In contrast, SGs formed in cells treated with NG conditions are not inhibited by ISRIB but can be modulated by an alternative intervention on the mTOR pathway, highlighting the specific features of these SGs.

The reviewer asks about specific issues related to mechanisms of SG persistence, canonical SGs, and SG disassembly (ABC). We have followed the suggestions raised in the major points section to try to address these issues.

Major points:

1. Authors should use more markers for SGs to demonstrate co-staining of PABP+ foci with other SG components. Do polyadenylated mRNAs present in these foci?

Response - We recognize that SG composition is highly variable. While polyadenylated mRNA is an obligate component of the majority of SGs (also shown in the current study – Fig. 6), SG protein composition is variable and dynamic, as well as stress and cell type context specific. Nonetheless, in the present study, we have used a total of four different markers for our *in vitro* studies, two well defined SG markers including G3BP1 and PABP and two other proteins reported as localizing to SGs in specific contexts (hnRNP A1 and TDP-43). We have also used

G3BP1 and PABP for our *in situ* studies. The *in vitro* data is provided in Fig. 2 and Fig. 6; the *in situ* data in Fig 1B, 1C and Fig. S1B. We accept that one could use multiple additional markers.

2. Puromycin/ cycloheximide treatments should be used to demonstrate that SG-like cytoplasmic foci are actually dynamic.

Response - We thank the reviewer for this suggestion. We have conducted the suggested experiments, supporting the conclusion that SG-like cytoplasmic foci are actually dynamic in our system. We have documented the impact of puromycin and cycloheximide (CHX) on SG formation in human primary OLs in Fig. 3D.

- The addition of puromycin to NG condition at 4 hours, the point at which more than half of the population contains SGs, resulted in increased SG formation. Our observations support previous mechanistic findings showing that polysome destabilization with puromycin enhances SG formation during stress.

- In contrast, CHX is known to stabilize polysome complexes and prevent SG assembly, perhaps dissolve existing ones. Our new data show that the addition of CHX to NG at 4 hours or to SA at 24 hours inhibits SG formation in OLs.

To emphasize the mechanistic aspect related to protein synthesis and polysome complexes equilibrium in our study, we have added comments in the Results and Discussion sections.

3. How pharmacological interventions (puromycin/ cycloheximide) with protein synthesis affect "persistent SGs"?

Response - We also performed these modulatory experiments using the cells treated with the combination of cytokines in NG conditions. We found that CHX molecule added to NG+TNF α +IFN γ conditions reduce the persistence of SGs at 24 hours (added to Fig 3D).

4. What is actual significance of PABP and hnRNPA1 coalescence with G3BP1-labeled SGs? Or the absence of TDP-43? What this mislocalization findings tell us?

Response - As mentioned in our response to review item #1, there can be multiple and variable compositions of SGs. Thus, we cannot specifically comment about the relevance of the specific combination of these markers. With regards to TDP-43 mislocalization, this has been previously associated with degeneration of OLs (Wang et al., 2018 – PMID: 30373824).

5. Do authors think that main observations of this work obtained in OLs are unique to OLs or it is common phenomenon for other cell types?

Response - In our Fig.S1E, we show that we can detect SGs in astrocytes *in situ* in MS lesions. We now refer to this in our Results section.

The control of protein synthesis and subsequent formation of SGs have been identified as key process to guide cellular survival in a large majority of eukaryote cells. It is a common mechanism shown in primary cells such as neurons (Salapa et al., 2018 – PMID: 30190085).

6. Finally, it is important to know what is actually happening in diseased state (MS) since majority of the experiments are done using human primary mature OLs.

Response - We agree with the reviewer on the importance of knowing what is actually happening in the disease state (MS) and for this reason, initiated our studies with analyses of MS tissues. The data in Fig. 1 and Fig S1B-S1D, are from our immunohistochemistry analyses of SG marker expression in OLs in active and inactive areas of MS lesions, covering the progressive evolution of the disorder. These results are supported by RNA-seq analyses using publicly available data bases (Fig. 7).

Reviewer #3 (Remarks to the Author):

The manuscript by Pernin investigates the role of stress granules in oligodendrocytes focusing on the pathological condition in multiple sclerosis (MS). Stress granules is well documented in many pathological conditions, but their functions, regulations, and therapeutic implications are under studied. Moreover, their presence in MS has never been intensively studied, therefore this manuscript has great importance to the field. The investigators first confirm using human multiple sclerosis brain samples the presence of stress granules. Comparison to the control brain tissues has provided clear evidence for the presence of stress granules in MS. The investigators then used in vitro model to demonstrate types of stressors that initiate the formation of stress granules in human oligodendrocytes. The use of human oligodendrocytes is another important piece of validation for stress granules. Lastly, the investigators confirm their finding using publicly available RNA-seq data from 3 independent studies. These analyses confirm the presence of stress granule related genes in MS tissues. Taken together, the investigators have gathered convincing evidence for the presence of stress granules in MS, which provides substantial advance to the field. The quality of each experiment is high, demonstrating replicability across publicly available datasets. Despite above mentioned strengths of the study, lack of critical information makes it difficult to assess the validity of some aspects of experiments. Investigators are requested to provide further information explained below, and also make modifications to the figures suggested below.

Major points:

1. Abstract – the word “ISIRB” need to be defined.

Response - We spell out the full name of ISRIB in the Introduction – it is widely known as ISRIB. We would prefer to use “*ISRIB*” in the abstract.

2. Fig 1 – the authors state that “Lesion activity was characterized by the presence of macrophages/microglia throughout the lesion area” to define MS regions. An independent figure is needed to demonstrate this assessment. Clarification is needed to show how investigators delineated NAWM, active, inactive areas.

Response - As mentioned for Reviewer 1 item #2, we now include the description of these two lesion categories (active vs inactive lesions) directly in the Results section: “*Lesion activity was characterized by LFB and H&E staining. Areas of ongoing demyelination were selected based on presence of macrophages/microglia, some of which contained LFB-positive material (Fig. S1A). There were also areas of inactivity in these active/mixed lesions.*”

We also provide a new supplementary figure of an active MS lesion (Fig.S1A) showing LFB-positive material in macrophages/microglia.

3. Fig 1 A/B – Nogo immunofluorescence in control WM seems lower in panel B than the actual quantification shown in panel A. Is this truly a representative image?

Response - Thank you for bringing this forward. We have reviewed our images and note that this was not representative. The image has been replaced with one that is more representative of WM (illustrating a higher NogoA signal)

4. Fig 1B/C – inactive area does not show PABP staining colocalizing with Nogo, in contrast to the quantification in panel C. Is this truly a representative image?

Response - We appreciate the feedback. Careful review of the images collected would suggest that a more representative image of this area is warranted. The image has been replaced with one showing PABP granules in NogoA positive cell.

5. Fig S1C – the demonstration of DAB staining does not show colocalization with oligodendrocyte marker(s) therefore does not add further validity to what is already stated in Fig1. It is recommended to remove this panel.

Response - We do consider that these images give additional support to our immunofluorescence staining from Fig.1B and Fig. S1B, showing SG formation in OLs in MS brain tissue. However, if the reviewer thinks this technique is not needed, we can remove this panel from the supplementary Figure 1.

6. Fig 1D – needs to increase the contrast for ATF4 fluorescence. It shows very dim, making it difficult to assess the claim.

Response - We have now boosted the signal intensity uniformly across all channels so as to maintain the relative signal intensity.

7. Fig 1F - needs to increase the contrast for p4E-BP1 fluorescence. It shows very dim, making it difficult to assess the claim.

Response - As above, we have adjusted the signal intensity across all channels.

8. Fig 2 and Fig 4 – the magenta color needs to be labeled as “O4”, and blue needs to be labeled “Dapi” within the panels, just like G3BP1 is labeled within the panels.

Response - We have now adapted all figures to fit with this method of labelling.

9. Fig 6C – it is recommended to increase the contrast of TDP-43. The fluorescent is dim and difficult to assess the claim.

Response - As above, we have adjusted the signal intensity across all channels.

10. Although Fig. 2 legend states N is ‘independent biological replicate’, additional explanation needs to be included regarding how quantification was made, such as the number of total cells/fields of interest/technical replicates quantified. This can be in the method section or in the legend.

Response - As stated in the method section: *“Data were derived by blinded observers counting >100 cells per time point, per condition and experiment.”*

11. Study design – There needs to be a comprehensive table with description of MS patient and control tissues including number of cases, age, type of MS, etc. Also needed is the explanation of type of surgically resected tissues (quantity, pathology, age, criteria for use in case of pathological tissues, etc). Throughout the manuscript, lack of these information makes it difficult to assess the replicability of findings.

Response - We thank the reviewer for this suggestion and agree that this is an important point. These data are now provided in Supplementary Table 1 and 2.

12. Human cell samples – the authors state “human OL cells culture in vitro is regularly checked by immunofluorescence using O4 antibody and estimated at >95% purity”. This needs to be demonstrated in an independent figure. Without the validation of OL culture purity and viability, it is difficult to assess the validity of in vitro experiments.

Response - We routinely check the purity of our OL cultures using immunofluorescence analyses, as stated in our previous publication (Pernin et al., 2022 – reference 15). From our recent studies that include the current data set, the mean % O4⁺ cells in dissociated cell cultures was 91.5±1.4%. We can add this number if requested.

We have additionally confirmed the purity of our culture via our bulk RNA seq analysis.

Reviewer #4 (Remarks to the Author):

Identifying the mechanisms mediating oligodendrocyte (OL) injury at distinct stages of MS represents a critical topic in the field with major therapeutic implications. Substantial prior evidence has suggested that both metabolic and classic inflammatory pathways likely contribute, particularly in chronic stages of MS. The work presented here addresses this question in an insightful manner, focusing on the relationship of stress granule (SG) formation to metabolic and inflammatory stressors. The authors convincingly demonstrate that SG formation occurs in OLs of MS patients, including NAWM and lesions of distinct pathologic stages, and they relate these findings to disruption of protein translation, energy deficiency, and cell death. They nicely demonstrate the distinct and yet synergistic effects of glucose deprivation and inflammatory cytokine stimulation, altogether providing important new insights into potential drivers of OL injury in MS. This is an important and relevant study. I have only a few minor comments to address:

Response - Thank you for the very positive assessment of our work.

Major points:

1. At the beginning of the results section, a brief description of the human MS and control tissue sample set would be helpful to understanding the data. Some details are provided in the Methods, but the readers' understanding would be improved if these were included in the main text. If possible, a simple supplementary table with these details would also be helpful. Important details include how many human subjects were included, age, sex, and disease duration (if known).

Response - As mentioned in response to Reviewer 3 item #11, we agree with the reviewers on this important point. These data are now provided in Supplementary Table 1 and 2.

2. In Figure 1, it seems surprising that the number of NogoA+ cells was only mildly decreased in active lesions compared to control WM and MS NAWM. This raises a couple questions.
- First, can more detail be given about how lesions were identified and classified? This was mentioned only briefly.

Response - The data on number of NogoA+ cells is very compatible with the previous report from Hess et al., 2020 (ref 19) that show that OL cell bodies are preserved in early active MS lesions and then decrease with lesion progression

As in our response for Reviewer 1 item #2 and Reviewer 3 item#2, we now include the description of these two lesion categories (active vs inactive lesions) directly in the Results section.

We also provide a new supplementary figure of an active MS lesion (Fig.S1A) showing LFB-positive material in macrophages/microglia.

- Was classification consistent with prior pathologic studies in human MS post-mortem samples?

Response - Yes, we based our histological analyses on the classification system suggested by Kuhlmann and colleagues (2017 – PMID: 27988845).

- Second, although NogoA is used as a marker of OLs, it is not exclusively expressed in OLs and has been reported in neurons among other cell types. Were NogoA+ cells clearly identified as OLs in this study? And how was this accomplished?

Response - In our previous and ongoing collaboration with Dr. Tanja Kuhlmann, we have used NogoA and TPPP antibodies to label mature OLs in brain tissue sections. These two markers were found to be overlapping (Kuhlmann et al., 2020 – PMID: 32710244).

3. With no glucose (NG), protein translation remains low at 24 hours despite SG formation being only transient (resolved by about 8 hours). This would suggest that SG formation is not the only determinant of translation in OLs. Can this be addressed in the text or discussion?

Response - As the reviewer points out, SGs are not the only regulator of protein synthesis and translation initiation. In line with this, our data in Fig. 3B show OLs with inhibition of protein synthesis at 24 hours with no SG formation. This supports that there are other determinants of protein translation under our stress conditions. We have included this observation in the Discussion section: *“As shown, OLs treated with NG conditions for 24 hours displayed sustained translation inhibition while no SGs were detected at this time. These observations are consistent with previous studies indicating that the mechanisms underlying SG persistence and those underlying inhibition of protein synthesis can be independent (reference 57- PMID: 30082464).”*

REVIEWERS' COMMENTS

Reviewer #1 (Remarks to the Author):

The revised manuscript from Pernin and colleagues is considerably improved. The authors have adequately addressed most points, with additional data supporting the claims. The manuscript is more clearly presented, and the data better support the claims. The work addresses an important area that will be of interest to a wide audience.

One concern uncovered in the evaluation of the revised manuscript has to do with relevant studies not discussed or mentioned. The authors may wish to perform a PUBMED search using terms such as "ISR and oligodendrocytes", "ISR and multiple sclerosis", "stress granules and oligodendrocytes", "inflammatory cytokines and multiple sclerosis", etc. A number of recent references, of apparent relevance, have not been cited or discussed.

Reviewer #2 (Remarks to the Author):

All my concerns were addressed adequately. I recommend this work for publication

Reviewer #3 (Remarks to the Author):

The revised manuscript has considered most of the critiques raised in my review, except for comment #5 (Fig S1C – the demonstration of DAB staining does not show colocalization with oligodendrocyte marker(s) therefore does not add further validity to what is already stated in Fig1. It is recommended to remove this panel.)

The authors still did not clarify how Fig S1C (now S1D) shows colocalization of PABP with oligodendrocytes without the use of oligodendrocyte marker. Because the reviewer cannot confirm the validity of the claim "Representative images of SGs (PABP+) in OLS"& "Arrows point SGs in OLS", it is requested the panel to be removed from the manuscript.

All other comments have been sufficiently addressed. I recommend the revised manuscript to be accepted after the issue above is addressed.

Reviewer #4 (Remarks to the Author):

The authors have adequately addressed my comments and those of the other reviewers. I believe this is an impactful piece of work that provides novel insights into oligodendrocyte responses to stressors that exist within MS lesions.

RESPONSES TO REVIWERS

Reviewer #1 (Remarks to the Author):

The revised manuscript from Pernin and colleagues is considerably improved. The authors have adequately addressed most points, with additional data supporting the claims. The manuscript is more clearly presented, and the data better support the claims. The work addresses an import area that will be of interest to a wide audience.

One concern uncovered in the evaluation of the revised manuscript has to do with relevant studies not discussed or mentioned. The authors may wish to perform a PUBMED search using terms such as “ISR and oligodendrocytes”, “ISR and multiple sclerosis”, “stress granules and oligodendrocytes”, “inflammatory cytokines and multiple sclerosis”, etc. A number of recent references, of apparent relevance, have not been cited or discussed.

Response - In accordance with this suggestion, we have added a select number of recent references on this topic (references 15, 59-60 and 67). We have had to limit the number to stay in line with the guidelines of the journal.

Reviewer #3 (Remarks to the Author):

The revised manuscript has considered most of the critiques raised in my review, except for comment #5 (Fig S1C – the demonstration of DAB staining does not show colocalization with oligodendrocyte marker(s) therefore does not add further validity to what is already stated in Fig1. It is recommended to remove this panel.)

The authors still did not clarify how Fig S1C (now S1D) shows colocalization of PABP with oligodendrocytes without the use of oligodendrocyte marker. Because the reviewer cannot confirm the validity of the claim "Representative images of SGs (PABP+) in OLs"& "Arrows point SGs in OLs", it is requested the panel to be removed from the manuscript.

All other comments have been sufficiently addressed. I recommend the revised manuscript to be accepted after the issue above is addressed.

Response - We have removed Figure S1D from the supplementary data, as requested.